# INFOrmation Prioritization through EmPOWERment in Visual Model-Based RL

**Homanga Bharadhwaj** [*]
Carnegie Mellon University

**Mohammad Babaeizadeh**
Google Research, Brain Team

**Dumitru Erhan**
Google Research, Brain Team

**Sergey Levine**
Google Research, Brain Team
University of California Berkeley

## Abstract

Model-based reinforcement learning (RL) algorithms designed for handling complex visual observations typically learn some sort of latent state representation, either explicitly or implicitly. Standard methods of this sort do not distinguish between functionally relevant aspects of the state and irrelevant distractors, instead aiming to represent all available information equally. We propose a modified objective for model-based RL that, in combination with mutual information maximization, allows us to learn representations and dynamics for visual model-based RL without reconstruction in a way that explicitly prioritizes functionally relevant factors. The key principle behind our design is to integrate a term inspired by variational empowerment into a state-space model based on mutual information. This term prioritizes information that is correlated with action, thus ensuring that functionally relevant factors are captured first. Furthermore, the same empowerment term also promotes faster exploration during the RL process, especially for sparse-reward tasks where the reward signal is insufficient to drive exploration in the early stages of learning. We evaluate the approach on a suite of vision-based robot control tasks with natural video backgrounds, and show that the proposed prioritized information objective outperforms state-of-the-art model based RL approaches with higher sample efficiency and episodic returns.

## 1 Introduction

Model-based reinforcement learning (RL) provides a promising approach to accelerating skill learning: by acquiring a predictive model that represents how the world works, an agent can quickly derive effective strategies, either by planning or by simulating synthetic experience under the model. However, in complex environments with high-dimensional observations (e.g., images), modeling the full observation space can present major challenges. While large neural network models have made progress on this problem (Finn & Levine, 2017; Ha & Schmidhuber, 2018; Hafner et al., 2019a; Watter et al., 2015; Babaeizadeh et al., 2017), effective learning in visually complex environments necessitates some mechanism for learning representations that *prioritize* functionally relevant factors for the current task. This needs to be done without wasting effort and capacity on *irrelevant* distractors, and without detailed reconstruction. Several recent works have proposed contrastive objectives that maximize mutual information between observations and latent states (Hjelm et al., 2018; Ma et al., 2020; Oord et al., 2018; Srinivas et al., 2020). While such objectives avoid reconstruction, they still do not distinguish between relevant and irrelevant factors of variation. We thus pose the question: can we devise non-reconstructive representation learning methods that explicitly prioritize information that is most likely to be functionally relevant to the agent?

In this work, we derive a model-based RL algorithm from a combination of representation learning via mutual information maximization (Poole et al., 2019) and empowerment (Mohamed & Rezende, 2015). The latter serves to drive both the representation and the policy toward exploring and representing functionally relevant factors of variation. By integrating an empowerment-based term into a

---

[*]Work done during Homanga's research internship at Google. `hbharadh@cs.cmu.edu`

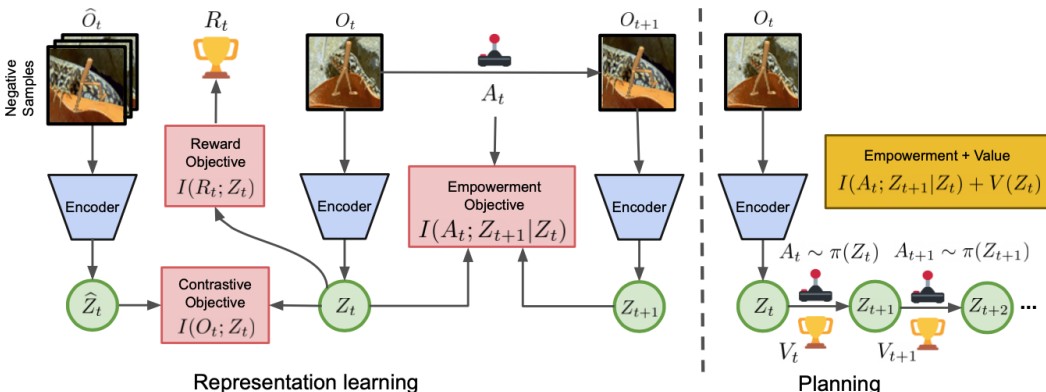

Figure 1: Overview of `InfoPower`. $I(\mathcal{O}_t; Z_t)$ is the contrastive learning objective for learning an encoder to map from image $\mathcal{O}$ to latent $Z$. $I(A_{t-1}; Z_t|Z_{t-1})$ is the empowerment objective that prioritizes encoding controllable representations in $Z$. $-I(i_{t+1}; Z_{t+1}|Z_t, A_t)$ helps learn a latent forward dynamics model so that future $Z_{t+k}$ can be predicted from current $Z_t$. $I(R_t; Z_t)$ helps learn a reward prediction model, such that the agent can learn a plan $A_t, ..A_{t+k}, ..$ through latent rollouts. Together, this combination of terms produces a latent state space model for MBRL that captures all necessary information at convergence, while prioritizing the most functionally relevant factors via the empowerment term.

mutual information framework for learning state representations, we effectively *prioritize* information that is most likely to have functional relevance, which mitigates distractions due to irrelevant factors of variation in the observations. By integrating this same term into policy learning, we further improve exploration, particularly in the early stages of learning in sparse-reward environments, where the reward signal provides comparatively little guidance.

Our main contribution is `InfoPower`, a model-based RL algorithm for control from image observations that integrates empowerment into a mutual information based, non-reconstructive framework for learning state space models. Our approach explicitly prioritizes information that is most likely to be functionally relevant, which significantly improves performance in the presence of time-correlated distractors (e.g., background videos), and also accelerates exploration in environments when the reward signal is weak. We evaluate the proposed objectives on a suite of simulated robotic control tasks with explicit video distractors, and demonstrate up to 20% better performance in terms of cumulative rewards at 1M environment interactions with 30% higher sample efficiency at 100k interactions.

## 2 PROBLEM STATEMENT AND NOTATION

A partially observed Markov decision process (POMDP) is a tuple $(\mathcal{S}, \mathcal{A}, T, R, \mathcal{O})$ that consists of states $s \in \mathcal{S}$, actions $a \in \mathcal{A}$, rewards $r \in R$, observations $o \in \mathcal{O}$, and a state-transition distribution $T(s'|s, a)$. In most practical settings, the agent interacting with the environment doesn't have access to the actual states in $\mathcal{S}$, but to some partial information in the form of observations $\mathcal{O}$. The underlying state-transition distribution $T$ and reward distribution $R$ are also unknown to the agent.

In this paper, we consider the observations $o \in \mathcal{O}$ to be high-dimensional images, and so the agent should learn a compact representation space $Z$ for the latent state-space model. The problem statement is to learn effective representations from observations $\mathcal{O}$ when there are visual distractors present in the scene, and plan using the learned representations to maximize the cumulative sum of discounted rewards, $J = \mathbb{E}[\sum_t \gamma^{t-1} r_t]$. The value of a state $V(Z_t)$ is defined as the expected cumulative sum of discounted rewards starting at state $Z_t$.

We use $q(\cdot)$ to denote parameterized variational approximations to learned distributions. We denote random variables with capital letters and use lowercase letters to denote particular realizations (e.g., $z_t$ denotes the value of $Z_t$). Since the underlying distributions are unknown, we evaluate all expectations through Monte-Carlo sampling with observed state-transition tuples $(o_t, a_{t-1}, o_{t-1}, z_t, z_{t-1}, r_t)$.

## 3 INFORMATION PRIORITIZATION FOR THE LATENT STATE-SPACE MODEL

Our goal is to learn a latent state-space model with a representation $Z$ that prioritizes capturing functionally relevant parts of observations $\mathcal{O}$, and devise a planning objective that explores with the learned representation. To achieve this, our key insight is integration of empowerment in the visual model-based RL pipeline. For representation learning we maximize MI $\max_Z I(\mathcal{O}, Z)$ subject to a prioritization of the empowerment objective $\max_Z I(A_{t-1}; Z_t | Z_{t-1})$. For planning, we maximize the empowerment objective along with reward-based value with respect to the policy $\max_A I(A_{t-1}; Z_t | Z_{t-1}) + I(R_t; Z_t)$.

In the subsequent sections, we elaborate on our approach, `InfoPower`, and describe lower bounds to MI that yield a tractable algorithm.

### 3.1 LEARNING CONTROLLABLE FACTORS AND PLANNING THROUGH EMPOWERMENT

Controllable representations are features of the observation that correspond to entities which the agent can influence through its actions. For example, in quadrupedal locomotion, this could include the joint positions, velocities, motor torques, and the configurations of any object in the environment that the robot can interact with. For robotic manipulation, it could include the joint actuators of the robot arm, and the configurations of objects in the scene that it can interact with. Such representations are denoted by $S^+$ in Fig. 2, which we can formally define through conditional independence as the smallest subspace of $S$, $S^+ \leq S$, such that $I(A_{t-1}; S_t | S_t^+) = 0$. This conditional independence relation can be seen in Fig. 2. We explicitly priori-

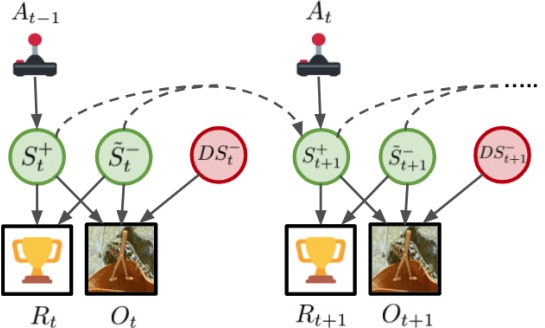

Figure 2: PGM showing decomposition of state $S$ into controllable parts $S^+$ (directly influenced by actions $A$), parts not influenced by actions that still influence the reward, $\tilde{S}^-$, and distractors $DS^-$. $\tilde{S}_{t+1}$ may be influenced by $\tilde{S}_t^-$ (arrow not shown to reduce clutter) but not by $\tilde{S}_t^+$.

tize the learning of such representations in the latent space by drawing inspiration from variational empowerment (Mohamed & Rezende, 2015).

The empowerment objective can be cast as maximizing a conditional information term $I(A_{t-1}; Z_t | Z_{t-1}) = \mathcal{H}(A_{t-1} | Z_{t-1}) - \mathcal{H}(A_{t-1} | Z_t, Z_{t-1})$. The first term $\mathcal{H}(A_{t-1} | Z_{t-1})$ encourages the chosen actions to be as diverse as possible, while the second term $-\mathcal{H}(A_{t-1} | Z_t, Z_{t-1})$ encourages the representations $Z_t$ and $Z_{t+1}$ to be such that the action $A_t$ for transition is predictable. While prior approaches have used empowerment in the model-free setting to learn policies by exploration through intrinsic motivation (Mohamed & Rezende, 2015), we specifically use this objective in combination with MI maximization for prioritizing the learning of controllable representations from distracting images in the latent state-space model.

We include the same empowerment objective in *both* representation learning and policy learning. For this, we augment the maximization of the latent value function that is standard for policy learning in visual model-based RL (Sutton, 1991), with $\max_A I(A_{t-1}; Z_t | Z_{t-1})$. This objectives complements value based-learning and further improves exploration by seeking controllable states. We empirically analyze the benefits of this in sections 4.3 and 4.5.

In Appendix A.1 we next describe two theorems regarding learning controllable representations. We observe that the $\max \sum_t I(A_{t-1}; Z_t | Z_{t-1})$ objective for learning latent representations $Z$, when used along with the planning objective, provably recovers controllable parts of the observation $\mathcal{O}$, namely $S^+$. This result in Theorem 1 is important because in practice, we may not be able to represent every possible factor of variation in a complex environment. In this situation, we would expect that when $|Z| \ll |\mathcal{O}|$, learning $Z$ under the objective $\max \sum_t I(A_{t-1}; Z_t | Z_{t-1})$ would encode $S^+$.

We further show through Theorem 2 that the inverse information objective alone can be used to train a latent-state space model and a policy through an alternating optimization algorithm that converges to a local minimum of the objective $\max \sum_t I(A_{t-1}; Z_t | Z_{t-1})$ at a rate inversely proportional to

the number of iterations. In Section 4.3 we empirically show how this objective helps achieve higher sample efficiency compared to pure value-based policy learning.

## 3.2 Mutual Information maximization for Representation Learning

For visual model-based RL, we need to learn a representation space $Z$, such that a forward dynamics model defining the probability of the next state in terms of the current state and the current action can be learned. The objective for this is $\sum_t -I(i_t; Z_t | Z_{t-1}, A_{t-1})$. Here, $i_t$ denotes the dataset indices that determine the observations $p(o_t|i_t) = \delta(o_t - o_{t'})$. In addition to the forward dynamics model, we need to learn a reward predictor by maximizing $\sum_t I(R_t; Z_t)$, such that the agent can plan ahead in the future by rolling forward latent states, without having to execute actions and observe rewards in the real environment.

Finally, we need to learn an encoder for encoding observations $\mathcal{O}$ to latents $Z$. Most successful prior works have used reconstruction-loss as a natural objective for learning this encoder (Babaeizadeh et al., 2017; Hafner et al., 2019b;a). A reconstruction-loss can be motivated by considering the objective $I(\mathcal{O}, Z)$ and computing its BA lower

---

**Algorithm 1:** Information Prioritization in Visual Model-based RL (`InfoPower`)

---

Initialize dataset $\mathcal{D}$ with random episodes.
Initialize model parameters $\phi, \chi, \psi, \eta$.
Initialize dual variable $\lambda$.
**while** *not converged* **do**
    **for** *update step* $c = 1..C$ **do**

        `// Model learning`
        Sample data $\{(a_t, o_t, r_t)\}_{t=k}^{k+L} \sim \mathcal{D}$.
        Compute latents $z_t \sim p_\phi(z_t | z_{t-1}, a_{t-1}, o_t)$.
        Calculate $\mathcal{L}$ based on section 3.4.
        $(\phi, \chi, \psi, \eta) \leftarrow (\phi, \chi, \psi, \eta) + \nabla_{\phi,\chi,\psi,\eta} \mathcal{L}$
        $\lambda \leftarrow \lambda - \nabla_\lambda \mathcal{L}$

        `// Behavior learning`
        Rollout latent plan, $\mathcal{S} \leftarrow \mathcal{S} \cup \{z_t, a_t, r_t\}$
        $V(z_t) \approx \mathbb{E}_\pi[\ln q_\eta(r_t|z_t) + \ln q_\psi(a_{t-1}|z_t, z_{t-1})]$
        Update policy $\pi$ and value model
    **end**

    `// Environment interaction`
    **for** *time step* $t = 0..T - 1$ **do**
        $z_t \sim p_\phi(z_t | z_{t-1}, a_{t-1}, o_t); a_t \sim \pi(a_t|z_t)$
        $r_t, o_{t+1} \leftarrow$ `env.step`$(a_t)$.
    **end**
    Add data $\mathcal{D} \leftarrow \mathcal{D} \cup \{(o_t, a_t, r_t)_{t=1}^T\}$.

**end**

---

bound (Agakov, 2004). $I(o_t; z_t) \geq \mathbb{E}_{p(o_t, z_t)}[\log q_{\phi'}(o_t|z_t)] + \mathcal{H}(p(o_t))$. The first term here is the reconstruction objective, with $q_{\phi'}(o_t|z_t)$ being the decoder, and the second term can be ignored as it doesn't depend on $Z$. However, this reconstruction objective explicitly encourages encoding the information from every pixel in the latent space (such that reconstructing the image is possible) and hence is prone to not ignoring distractors.

In contrast, if we consider other lower bounds to $I(\mathcal{O}, Z)$, we can obtain tractable objectives that do not involve reconstructing high-dimensional images. We can obtain an *NCE*-based lower bound (Hjelm et al., 2018): $I(o_t; z_t) \geq \mathbb{E}_{q_\phi(z_t|o_t)p(o_t)}[\log f_\theta(z_t, o_t) - \log \sum_{t' \neq t} f_\theta(z_t, o_{t'})]$, where $q_\phi(z_t|o_t)$ is the learned encoder, $o_t$ is the observation at timestep $t$ (positive sample), and all observations in the replay buffer $o_{t'}$ are negative samples. $f_\theta(z_t, o_{t'}) = \exp(z_t^T W_\theta z_{t'})$ The lower-bound is a form of contrastive learning as it maximizes compatibility of $z_t$ with the corresponding observation $o_t$ while minimizing compatibility with all other observations across time and batch.

Although prior work has explored NCE-based bounds for contrastive learning in RL (Srinivas et al., 2020), to the best of our knowledge, prior work has not used this in conjunction with empowerment for prioritizing information in visual model-based RL. Similarly, the *Nguyen-Wainwright-Jordan (NWJ)* bound (Nguyen et al., 2010), which to the best our knowledge has not been used by prior works in visual model-based RL, can be obtained as,

$$I(o_t; z_t) \geq \mathbb{E}_{q_\phi(z_t|o_t)p(o_t)}[f_\theta(z_t, o_t)] - e^{-1}\mathbb{E}_{q_\phi(z_t|o_t)p(o_t)}e^{f_\theta(z_t, o_t)},$$

where $f_\theta$ is a *critic*. There exists an optimal critic function for which the bound is tightest and equality holds.

We refer to the InfoNCE and NWJ lower bound based objectives as *contrastive learning*, in order to distinguish them from a reconstruction-loss based objective, though both are bounds on mutual

information. We denote a lower bound to MI by $\underline{I(o_t, z_t)}$. We empirically find the NWJ-bound to perform slightly better than the NCE-bound for our approach, explained in section 4.5.

### 3.3 OVERALL OBJECTIVE

We now motivate the overall objective, which consists of maximizing mutual information while prioritizing the learning of controllable representations through empowerment in a latent state-space model. Based on the discussions in Sections 3.1 and 3.2, we define the overall objective for *representation learning* as

$$\max_{Z_{0:H-1}} \sum_{t=0}^{H-1} I(\mathcal{O}_t; Z_t) \text{ s.t. } \sum_{t=0}^{H-1} \overbrace{(-I(i_t; Z_t|Z_{t-1}, A_{t-1}) + I(A_{t-1}; Z_t|Z_{t-1}) + I(R_t; Z_t))}^{\mathcal{C}_t} \geq c_0.$$

The objective is to maximize a MI term $I(\mathcal{O}_t; Z_t)$ through contrastive learning such that a constraint on $\mathcal{C}_t$ holds for prioritizing the encoding of forward-predictive, reward-predictive and controllable representations. We define the overall *planning* objective as

$$\max_{A_{0:H-1}} \sum_{t=0}^{H-1} I(A_{t-1}; Z_t|Z_{t-1}) + V(Z_t) \quad ; A_t = \pi(Z_t) \quad ; V(Z_t) \approx \sum_t R_t.$$

The planning objective is to learn a policy as a function of the latent state $Z$ such that the empowerment term and a reward-based value term are maximized over the horizon $H$.

We can perform the constrained optimization for representation learning through the method of Lagrange Multipliers, by the primal and dual updates shown in Section A.2. In order to analyze this objective, let $|\mathcal{O}| = n$ and $|Z| = d$. Since, $\mathcal{O}$ corresponds to images and $Z$ is a bottlenecked latent representation, $d \ll n$.

$I(\mathcal{O}, Z)$ is maximized when $Z$ contains all the information present in $\mathcal{O}$ such that $Z$ is a sufficient statistic of $\mathcal{O}$. However, in practice, this is not possible because $|Z| \ll |\mathcal{O}|$. When $c_0$ is sufficiently large, and the constraint $\sum_t \mathcal{C}_t \geq c_0$ is satisfied, $Z = [S^+, \tilde{S}^-]$. Hence, the objective $\max \sum_t I(\mathcal{O}_t, Z_t)$ s.t. $\sum_t \mathcal{C}_t \geq c_0$ cannot encode anything else in $Z$, in particular it cannot encode distractors $DS^-$.

To understand the importance of prioritization through $\mathcal{C}_t \geq c_0$, consider $\max \sum_t I(\mathcal{O}_t, Z_t)$ without the constraint. This objective would try to make $Z$ a sufficient statistic of $\mathcal{O}$, but since $|Z| \ll |\mathcal{O}|$, there are no guarantees about which parts of $\mathcal{O}$ are getting encoded in $Z$. This is because both distractors $S^-$ and non-distractors $S^+, D\tilde{S}^-$ are equally important with respect to $I(\mathcal{O}, Z)$. Hence, the constraint helps in prioritizing the type of information to be encoded in $Z$.

### 3.4 PRACTICAL ALGORITHM AND IMPLEMENTATION DETAILS

To arrive at a practical algorithm, we optimize the overall objective in section 3.3 through lower bounds on each of the MI terms. For $I(\mathcal{O}, Z)$ we consider two variants, corresponding to the NCE and NWJ lower bounds described in Section 3.2. We can obtain a variational lower bound on each of the terms in $\mathcal{C}_t$ as follows, with detailed derivations in Appendix A.3:

$$-I(i_t; z_t|a_{t-1}) \geq -\sum_t \mathbb{E}[D_{\text{KL}}(p(z_t|z_{t-1}, a_{t-1}, o_t)||q_\chi(z_t|z_{t-1}, a_{t-1}))]$$

$$I(r_t; z_t) \geq \mathbb{E}_{p(r_t|o_t)}[\log q_\eta(r_t|z_t)] + \mathcal{H}(p(r_t))$$

$$I(a_{t-1}; z_t|z_{t-1}) \geq \mathbb{E}_{p(o_t|z_{t-1}, a_{t-1})q_\phi(z_t|o_t)}[\log q_\psi(a_{t-1}|z_t, z_{t-1})] + \mathbb{E}[\mathcal{H}(\pi(a_{t-1}|z_{t-1}))]$$

Here, $q_\phi(z_t|o_t)$ is the observation encoder, $q_\psi(a_{t-1}|z_t, z_{t-1})$ is the inverse dynamics model, $q_\eta(r_t|z_t)$ is the reward decoder, and $q_\chi(z_t|z_{t-1}, a_{t-1}))$ is the forward dynamics model. The inverse model helps in learning representations such that the chosen action is predictable from successive latent states. The forward dynamics model helps in predicting the next latent state given the current latent state and the current action, without having the next observation. The reward decoder predicts the reward (a scalar) at a time-step given the corresponding latent state. We use dense networks for all the models, with details in Appendix A.7. We denote by $\underline{\mathcal{C}_t}$, the lower bound to $\mathcal{C}_t$ based on the sum of the terms above. We construct a Lagrangian $\mathcal{L} = \sum_t \underline{I(o_t; z_t)} + \lambda (\underline{\mathcal{C}_t} - c_0)$ and optimize it by primal-dual gradient descent. An outline of this is shown in Algorithm 1.

**Planning Objective.** For planning to choose actions at every time-step, we learn a policy $\pi(a|z)$ through value estimates of task reward and the empowerment objective. We learn value estimates with $V(z_t) \approx \mathbb{E}_\pi[\ln q_\eta(r_t|z_t) + \ln q_\psi(a_{t-1}|z_t, z_{t-1})]$. We estimate $V(z_t)$ similar to Equation 6 of (Hafner et al., 2019a). The empowerment term $q_\psi(a_{t-1}|z_t, z_{t-1})$ in policy learning incentivizes choosing actions $a_{t-1}$ such that they can be predicted from consecutive latent states $z_{t-1}, z_t$. This biases the policy to explore controllable regions of the state-space. The policy is trained to maximize the estimate of the value, while the value model is trained to fit the estimate of the value that changes as the policy is updated.

Finally, we note that the difference in value function of the underlying MDP $Q^\pi(o, a)$ and the latent MDP $\hat{Q}^\pi(z, a)$, where $z \sim q_\phi(z|o)$ is bounded, under some regularity assumptions. We provide this result in Theorem 3 of the Appendix Section A.4. The overall procedure for model learning, planning, and interacting with the environment is outlined in Algorithm 1.

## 4 EXPERIMENTS

Through our experiments, we aim to understand the following research questions:

1. How does `InfoPower` compare with the baselines in terms of episodic returns in environments with explicit background video distractors?
2. How sample efficient is `InfoPower` when the reward signal is weak ($< 100k$ env steps when the learned policy typically doesn't achieve very high rewards)?
3. How does `InfoPower` compare with baselines in terms of behavioral similarity of latent states?

Please refer to the website for a summary and qualitative visualization results https://sites.google.com/view/information-empowerment

### 4.1 SETUP

We perform experiments with modified DeepMind Control Suite environments (Tassa et al., 2018), with natural video distractors from ILSVRC dataset (Russakovsky et al., 2015) in the background. The agent receives only image observations at each time-step and does not receive the ground-truth simulator state. This is a very challenging setting, because the agents must learn to ignore the distractors and abstract out representations necessary for control. While natural videos that are unrelated to the task might be easy to ignore, realistic scenes might have other elements that resemble the controllable elements, but are not actually controllable (e.g., other cars in a driving scenario). To emulate this, we also add distractors that resemble other potentially controllable robots, as shown for example in Fig. 3 (1st and 6th), but are not actually influenced by actions.

In addition to this setting, we also perform experiments with gray-scale distractors based on videos from the Kinetics dataset (Kay et al., 2017). This setting is adapted exactly based on prior works (Fu et al., 2021; Zhang et al., 2020) and we compare directly to the published results.

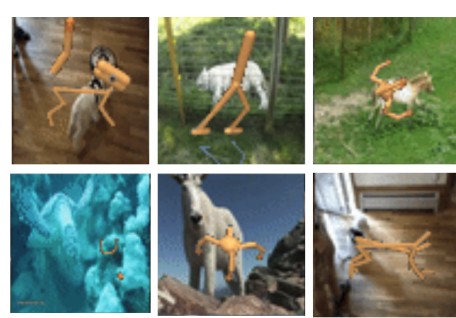

We compare `InfoPower` with state-of-the art baselines that also learn world models for control: *Dreamer* (Hafner et al., 2019a), *C-Dreamer* that is a contrastive version of Dreamer similar to Ma et al. (2020), *TIA* (Fu et al., 2021) that learns explicit reconstructions for both the distracting background and agent separately, *DBC* (Zhang et al., 2020), and *DeepMDP* (Gelada et al., 2019). We also compare variants of `InfoPower` with different MI bounds (NCE and NWJ), and ablations of it that remove the empowerment objective from policy learning.

Figure 3: Some of the natural video background distractors in our experiments. The videos change every 50 time-steps. Some backgrounds (for e.g. the top left and the bottom right) have more complex distractors in the form of agent-behind-agent i.e. the background has pre-recorded motion of a similar agent that is being controlled.

### 4.2 A BEHAVIORAL SIMILARITY METRIC BASED ON GRAPH KERNELS

In order to measure how good the learned latent representations are at capturing the functionally relevant elements in the scene, we introduce a behavioral similarity metric with details in A.5. Given

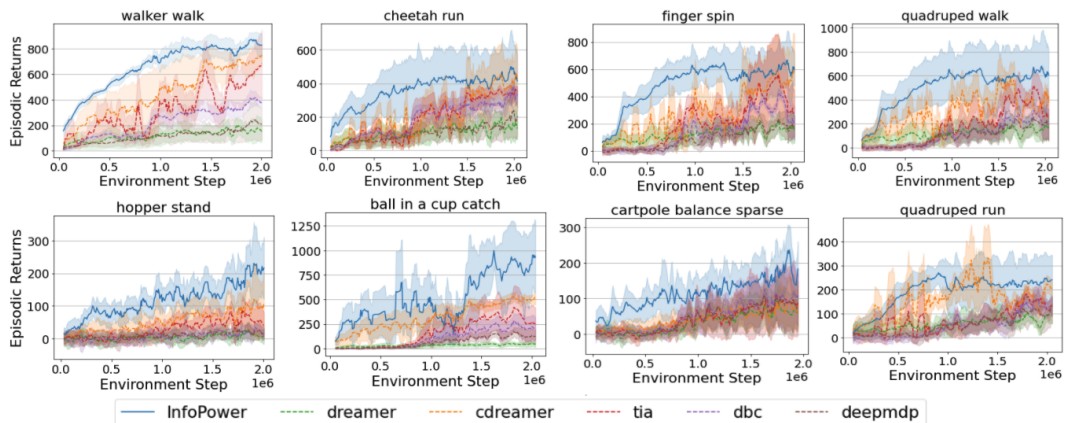

Figure 4: Evaluation of `InfoPower` and baselines in a suite of DeepMind Control tasks with natural video distractors in the background. The x-axis denotes the number of environment interactions and the y-axis shows the episodic returns. The S.D is over 4 random seeds. Higher is better.

the sets $\mathcal{S}_z = \{z^i\}_{i=1}^n$ and $\mathcal{S}_{z_{gt}} = \{z^i_{gt}\}_{i=1}^n$, we construct complete graphs by connecting every vertex with every other vertex through an edge. The weight of an edge is the Euclidean distance between the respective pairs of vertices. The label of each vertex is an unique integer and corresponding vertices in the two graphs are labelled similarly.

Let the resulting graphs be denoted as $G_z = (V_z, E_z)$ and $G_{z_{gt}} = (V_{z_{gt}}, E_{z_{gt}})$ respectively. Note that both these graphs are *shortest path* graphs, by construction. Let, $e_i = \{u_i, v_i\}$ and $e_j = \{u_j, v_j\}$. We define a kernel $\hat{k}$ that measures similarity between the edges $e_i, e_j$ and the labels on respective vertices. $\hat{k}(e_i, e_j) =$

$k_v(l(u_i), l(u_j)) k_e(l(e_i), l(e_j)) k_v(l(v_i), l(v_j)) + k_v(l(u_i), l(v_j)) k_e(l(e_i), l(e_j)) k_v(l(v_i), l(u_j))$

Here, the function $l(\cdot)$ denotes the labels of vertices and the weights of edges. $k_v$ is a Dirac delta kernel i.e. $k_e(x, y) = 1 - \delta(x, y)$ (Note that $\delta(x, y) = 1$ iff $x = y$, else $\delta(x, y) = 0$) and $k_e$ is a Brownian ridge kernel, $k_e(x, y) = \frac{1}{c} \max(0, c - |x - y|)$, where $c$ is a large number. We define the shortest path kernel to measure similarity between the two graphs, as, $k(G_z, G_{z_{gt}}) = \frac{1}{|E_z|} \sum_{e_i \in E_z} \sum_{e_j \in E_{z_{gt}}} \hat{k}(e_i, e_j)$. The value of $k(G_z, G_{z_{gt}})$ is low when a large number of pairs of corresponding vertices in both the graphs have the same edge length. We expect methods that recover latent state representations which better reflect the true underlying simulator state (i.e., the positions of the joints) to have higher values according to this metric.

## 4.3 SAMPLE EFFICIENCY ANALYSIS

In Fig. 4, we compare `InfoPower` and the baselines in terms of episodic returns. This version of `InfoPower` corresponds to an NWJ bound on MI which we find works slightly better than the NCE bound variant analyzed in section 4.5. It is evident that `InfoPower` achieves higher returns before 1M steps of training quickly compared to the baselines, indicating higher sample efficiency. This suggests the effectiveness of the empowerment model in helping capture controllable representations early on during training, when the agent doesn't take on actions that yield very high rewards.

## 4.4 BEHAVIORAL SIMILARITY OF LATENT STATES

In this section, we analyze how similar are the learned latent representations with respect to the underlying simulator states. The intuition for this comparison is that the proprioceptive features in the simulator state abstract out distractors in the image, and so we want the latent states to be behaviorally similar to the simulator states.

**Quantitative results with the defined metric.** In Table 1, we show results for behavioral similarity of latent states (Sim), based on the metric in section 4.2. We see that the value of Sim for `InfoPower` is around 20% higher than the most competitive baseline, indicating high behavioral similarity of the latent states with respect to the corresponding ground-truth simulator states.

**Qualitative visualizations with t-sne.** Fig. 5 shows a t-SNE plot of latent states $z \sim q_\phi(z|o)$ for `InfoPower` and the baselines with visualizations of 3 nearest neighbors for two randomly chosen latent states. We see that the state of the agent is similar is each group for `InfoPower`, although the

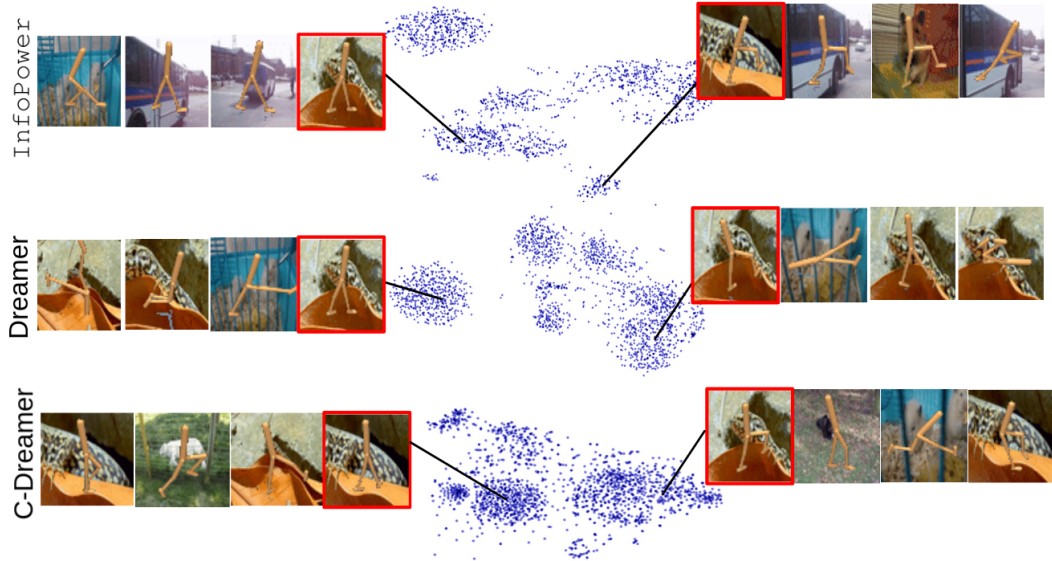

Figure 5: t-SNE plot of latent states $z \sim q_\phi(z|o)$ with visualizations of three nearest neighbors for two randomly sampled points (in red frame). We see that the state of the agent is similar is each set for `InfoPower`, whereas for Dreamer, and the most competitive baseline C-Dreamer, the nearest neighbor frames have significantly different agent configurations.

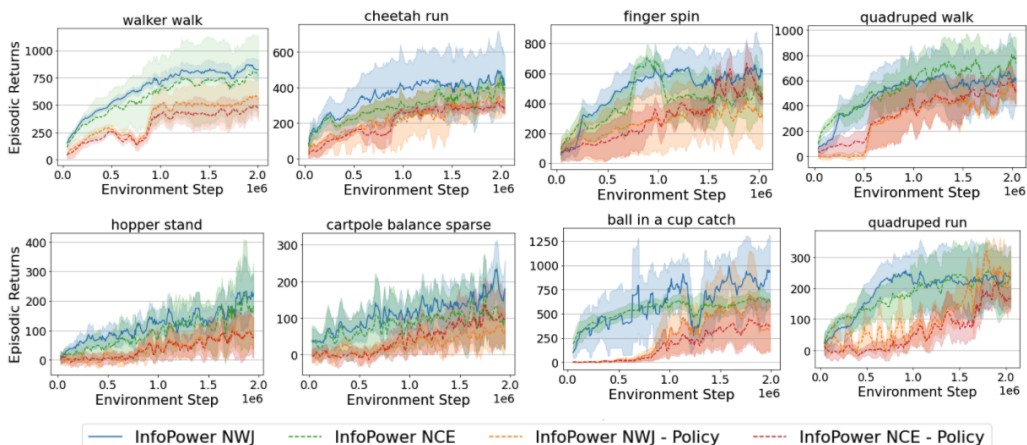

Figure 6: Evaluation of `InfoPower` and ablated variants in a suite of DeepMind Control tasks with natural video distractors in the background. The x-axis denotes the number of environment interactions and the y-axis shows the episodic returns. `InfoPower`-NWJ and `InfoPower`-NCE are full versions of our method differing only in the MI lower bound. The versions with - Policy do not include the empowerment objective in policy learning, but only use it from representation learning. The S.D is over 4 random seeds. Higher is better.

background scenes are significantly different. However, for the baselines, the nearest neighbor states are significantly different in terms of the pose of the agent, indicating that the latent representations encode significant background information.

## 4.5 ABLATION STUDIES

In Fig. 6, we compare different ablations of `InfoPower`. Keeping everything else the same, and changing only the MI lower bound to NCE, we see that the performance is almost similar or slightly worse. However, when we remove the empowerment objective from policy optimization (the versions with '-Policy' in the plot), we see that performance drops. The drop is significant in the regions $< 200k$ environment interactions, particularly in the sparse reward environments - cartpole balance and ball in a cup, indicating the necessity of the empowerment objective in exploration for learning controllable representations, when the reward signal is weak.

## 5 RELATED WORKS

**Visual model-based RL.** Recent developments in video prediction and contrastive learning have enabled learning of world-models from images (Watter et al., 2015; Babaeizadeh et al., 2017; Finn & Levine, 2017; Hafner et al., 2019a; Ha & Schmidhuber, 2018; Hafner et al., 2019b; Xie et al., 2020). All of these approaches learn latent representations through reconstruction objectives that are amenable for planning. Other approaches have used similar reconstruction based objectives for control, but not for MBRL (Lee et al., 2019; Gregor et al., 2019).

**MI for representation learning.** Mutual Information measures the dependence between two random variables. The task of learning latent representations $Z$ from images $\mathcal{O}$ for downstream applications, has been very successful with MI objectives of the form $\max_{f_1, f_2} I(f_1(\mathcal{O}), f_2(Z))$ (Hjelm & Bachman, 2020; Tian et al., 2020; Oord et al., 2018; Tschannen et al., 2019; Nachum & Yang, 2021). Since calculating MI exactly is intractable optimizing MI based objectives, it is important to construct appropriate MI estimators that lower-bound the true MI objective (Hjelm et al., 2018; Nguyen et al., 2010; Belghazi et al., 2018; Agakov, 2004). The choice of the estimator is crucial, as shown by recent works (Poole et al., 2019), and different estimators yield very different behaviors of the algorithm. We incorporate MI maximization through the NCE (Hjelm et al., 2018) and NWJ (Nguyen et al., 2010) lower bounds, such that typical reconstruction objectives for representation learning which do not perform well with visual distractors, can be avoided.

**Inverse models and empowerment.** Prior approaches have used inverse dynamics models for regularization in representation learning (Agrawal et al., 2016; Zhang et al., 2018) and as bonuses for improving policy gradient updates in RL (Shelhamer et al., 2016; Pathak et al., 2017). The importance of information theoretic approaches for learning representations that maximize predictive power has been discussed in prior work (Still, 2009) and more recently in Lee et al. (2020). Empowerment (Mohamed & Rezende, 2015) has been used as exploration bonuses for policy learning (Leibfried et al., 2019; Klyubin et al., 2008), and for learning skills in RL (Gregor et al., 2016; Sharma et al., 2019; Eysenbach et al., 2018). In contrast to prior work, we incorporate empowerment both for state space representation learning and policy learning, in a visual model-based RL framework, with the aim of prioritizing the most functionally relevant information in the scene.

**RL with environment distractors.** Some recent RL frameworks have studied the problem of abstracting out only the task relevant information from the environment when there are explicit distractors (Hansen & Wang, 2020; Zhang et al., 2020; Fu et al., 2021; Ma et al., 2020). Zhang et al. (2020) constrain the latent states by enforcing a bisimulation metric, without a reconstruction objective. Our approach is different from this primarily because of the empowerment objective that provides useful signal for dealiasing controllable vs. uncontrollable representations independent of how strong is the observed reward signal. Fu et al. (2021) model both the relevant and irrelevant aspects of the environment separately, and differ from our approach that prioritizes learning only the relevant aspects. (Shu et al., 2020) was and earlier approach that used contrastive representations for control. More recently, (Nguyen et al., 2021) used temporal predictive coding and (Ma et al., 2020) used InfoNCE based contrastive loss for learning representations while ignoring distractors in visual MBRL. Incorporating data augmentations for improving robustness with respect to environment variations is an orthogonal line of work (Laskin et al., 2020; Hansen & Wang, 2020; Raileanu et al., 2020; Srinivas et al., 2020; Kostrikov et al., 2020), complementary to our approach.

## 6 CONCLUSION

In this paper we derived an approach for visual model-based RL such that an agent can learn a latent state-space model and a policy by explicitly prioritizing the encoding of functionally relevant factors. Our prioritized information objective integrates a term inspired by variational empowerment into a non-reconstructive objective for learning state space models. We evaluate our approach on a suite of vision-based robot control tasks with two sets of challenging video distractor backgrounds. In comparison to state-of-the-art visual model-based RL methods, we observe higher sample efficiency and episodic returns across different environments and distractor settings.

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

# A APPENDIX

## A.1 THEORETICAL RESULTS ON CONTROLLABLE REPRESENTATIONS

Let $\Phi(\cdot)$ denote the encoder such that $Z = \Phi(\mathcal{O})$. $\mathcal{O}$ is the observation seen by the agent and $S$ is the underlying state. $S^+$ is defined as that part of underlying state $S$ which is directly influenced by actions $A$ i.e. $S^+ \leq S$ s.t. $I(A_{t-1}; S_t | S_t^+) = 0$.

We next describe two theorems regarding learning controllable representations, with proofs in the Appendix. We observe that the $\max \sum_t I(A_{t-1}; Z_t | Z_{t-1})$ objective alone for learning latent representations $Z$, along with the planning objective provably recovers controllable parts of the observation $\mathcal{O}$, namely $S^+$.

**Theorem 1.** *The objective* $\max \sum_t I(A_{t-1}; Z_t | Z_{t-1})$ *provably recovers controllable parts* $S^+$ *of the observation* $\mathcal{O}$. $S^+$ *is defined as that part of underlying state* $S$ *which is directly influenced by actions* $A$ *i.e.* $S^+ \subset S$ *s.t.* $I(S_t; A_{t-1} | S_t^+) = 0$.

*Proof.* Based on Fig. 2 and Fig. 7, we have the following

$$\max I(A_{t-1}; Z_t) \leq \max I(A_{t-1}; \mathcal{O}_t) \quad \text{Data-Processing Inequality}$$
$$\leq I(A_{t-1}; S_t) \quad \text{Data-Processing Inequality}$$
$$\leq I(A_{t-1}; S_t, S_t^+)$$
$$\leq I(A_{t-1}; S_t | S_t^+) + I(A_{t-1}; S_t^+) \quad \text{Chain Rule of MI}$$
$$\leq I(A_{t-1}; S_t^+) \quad \text{Conditional independence}$$

So, the maximum above can be obtained when the encoder $\Phi$ is an identity function over $S^+$ and a zero function over $[\tilde{S}^-, S^-]$. So, $Z_t = S_t^+$. Hence, given $Z_{t-1}$, $I(A_{t-1}; Z_t | Z_{t-1})$ is maximized when $\Phi$ is an identity function over $S^+$. So, $\max \sum_t I(A_{t-1}; Z_t | Z_{t-1})$ learns the encoder $\Phi$ such that controllable representations are recovered.

$\square$

This result is important because in practice, we may not be able to represent every possible factor of variation in a complex environment. In this situation, we would expect that when $|Z| \ll |\mathcal{O}|$, learning $Z$ under the objective $\max \sum_t I(A_{t-1}; Z_t | Z_{t-1})$ would encode $S^+$. We next show that the inverse information objective alone can be used to train a latent-state space model and a policy through an alternating optimization algorithm that converges to a local minimum of the objective $\max \sum_t I(A_{t-1}; Z_t | Z_{t-1})$ at a rate inversely proportional to the number of iterations.

**Theorem 2.** $\max_{\pi,\psi} \sum_t I(A_{t-1}; Z_t | Z_{t-1}) = \sum_t \mathbb{E}_{\pi(a_{t-1}|z_{t-1})p(z_t|z_{t-1},a_{t-1},o_t)} \log \frac{q_\psi(a_{t-1}|z_t,z_{t-1})}{\pi(a_{t-1}|z_{t-1})}$ *can be optimized through an alternating optimization scheme that has a convergence rate of* $\mathcal{O}(1/N)$ *to a local minima of the objective, where* $N$ *is the number of iterations.*

*Proof.* Let $z \sim q_\phi(z|o)$ denote a latent state sampled from the encoder distribution. Learning a world model such that reward prediction and inverse information are maximized can be summarized as:

$$\max_{\phi,\psi,\eta} \sum_t \mathbb{E}_{\pi(a_t|z_t),q_\phi(z_t|o_t)} \left[ \mathbb{E}_{p(r_t|z_t,a_t)} \log q_\eta(r_t|z_t,a_t) + \mathbb{E}_{p(z_{t+1}|z_t,a_t)} \log \frac{q_\psi(a_t|z_{t+1},z_t)}{\pi(a_t|z_t)} \right]$$

The policy $\pi(a_t|z_t)$ is learned by

$$\max_{\pi} \sum_t \mathbb{E}_{\pi(a_t|z_t),q_\phi(z_t|o_t)} \left[ \mathbb{E}_{p(r_t|z_t,a_t)} \log q_\eta(r_t|z_t,a_t) + \mathbb{E}_{p(z_{t+1}|z_t,a_t)} \log \frac{q_\psi(a_t|z_{t+1},z_t)}{\pi(a_t|z_t)} \right]$$

Model-based RL in this case involves alternating optimization between policy learning and learning the world model. In the case where there is no reward signal from the environment, i.e. $p(r_t|z_t)$, we have the following objective

$$\max_{\pi} \max_{\phi,\psi} \sum_t \mathbb{E}_{\pi(a_t|z_t),q_\phi(z_t|s_t)p(z_{t+1}|z_t,a_t)} \log \frac{q_\psi(a_t|z_{t+1},z_t)}{\pi(a_t|z_t)}$$

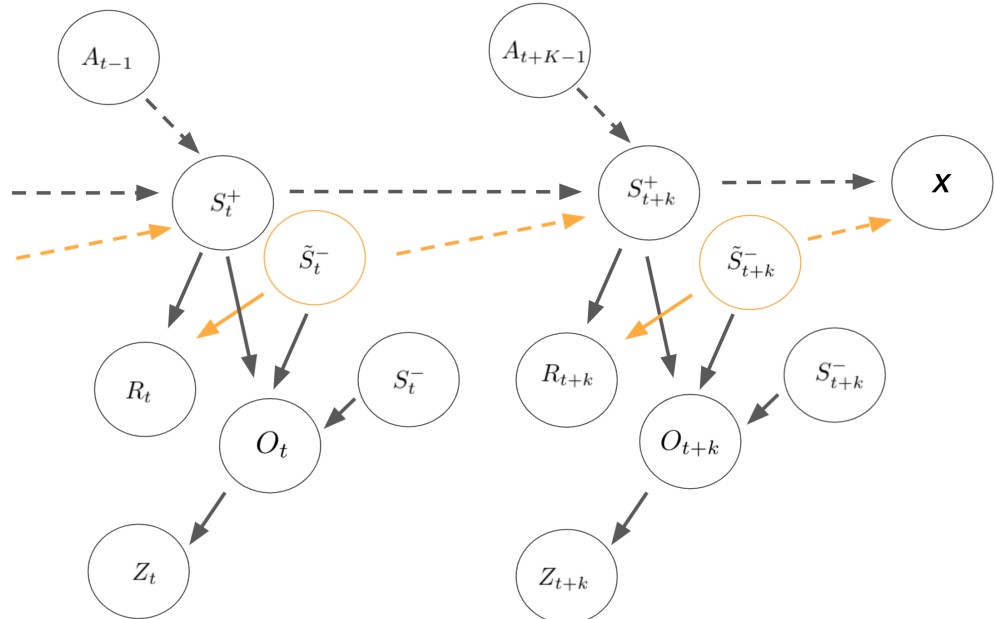

Figure 7: PGM of the MDP with distractor states. The state observed by the agent $\mathcal{O}$ consists of three parts $S^+$, $\tilde{S}^-$ and $DS^-$. $S^+$ is the controllable part of the state i.e. it is affected by the actions of the agent, and in turn affects the reward $R$; $\tilde{S}^-$ is not controllable by the agent but affects the reward $R$ and future $S^+$; $DS^-$ is not controllable by the agent and doesn't affect the reward $R$ and future $S^+$.

The optimal $q^*_\psi$ give a fixed $\pi$ and $\phi$ can be derived as in page 335 of Cover & Thomas (2006).

$$q^*_\psi(a_t|z_{t+1}, z_t) = \frac{q_\phi(z_t|s_t)p(z_{t+1}|z_t, a_t)\pi(a_t|z_t)}{\int_a q_\phi(z_t|s_t)p(z_{t+1}|z_t, a_t)\pi(a_t|z_t)}$$

The optimal $\pi*$ given a fixed $\psi$ and $\phi$ can be derived as

$$\pi^*(a_t|z_t) = \frac{\exp\left(\mathbb{E}_{q_\phi(z_t|s_t)p(z_{t+1}|z_t,a_t)}[\log q_\psi(a_t|z_{t+1}, z_t)]\right)}{\int_a \exp\left(\mathbb{E}_{q_\phi(z_t|s_t)p(z_{t+1}|z_t,a_t)q}[\log q_\psi(a_t|z_{t+1}, z_t)]\right)}$$

We note that these alternating iterations correspond to an instantiation of the Blahut-Arimoto algorithm (Blahut, 1972) for determining the information theoretic capacity of a channel. Based on (Nakagawa et al., 2021) we see that this iterative optimization procedure between $q^*_\psi$ and $\pi^*$ converges, and the worst-case rate of convergence is $\mathcal{O}(1/N)$ where $N$ is the number of iterations.

$\square$

This result is useful because it shows that even in the absence of rewards from the environment ($r_t = 0\ \forall t$), when planning to minimize regret is not possible, the inverse information objective can be used to train a policy that explores to seek out controllable parts of the state-space. When rewards from the environment are present, we can train the policy with this objective and the value estimates obtained from the cumulative rewards during planning. In Section 4.3 we empirically show how this objective helps achieve higher sample efficiency compared to pure value-based policy learning.

## A.2 PRIMAL DUAL UPDATES

$$Z_{0:H-1} = Z_{0:H-1} + \frac{\partial \sum_{t=0}^{H-1} I(\mathcal{O}_t; Z_t) + \lambda\left(-I(i_t; Z_t|Z_{t-1}, A_{t-1}) + I(A_{t-1}; Z_t|Z_{t-1}) + I(R_t; Z_t) - c_0\right)}{\partial Z_{0:H-1}}$$

$$\lambda = \lambda - \left( \underbrace{-I(i_t; Z_t|Z_{t-1}, A_{t-1})}_{\text{fwd dynamics model}} + \underbrace{I(A_{t-1}; Z_t|Z_{t-1})}_{\text{inv dynamics model}} + \underbrace{I(R_t; Z_t)}_{\text{rew model}} - c_0 \right)$$

## A.3 Forward, Inverse, and Reward Models

$$
\begin{aligned}
-I(i_t; Z_t|Z_{t-1}, A_{t-1}) &= -\int p(z_t, z_{t-1}, a_{t-1}, i_t, o_t) \log \frac{p(z_t|z_{t-1}, a_{t-1}, o_t)}{p(z_t|z_{t-1}, a_{t-1})} \\
&= -\int p(z_t, z_{t-1}, a_{t-1}, o_t) \left( \log \frac{p(z_t|z_{t-1}, a_{t-1}, o_t)}{q_\chi(z_t|z_{t-1}, a_{t-1})} + \log \frac{q_\chi(z_t|z_{t-1}, a_{t-1})}{p(z_t|z_{t-1}, a_{t-1})} \right) \\
&\geq -\int p(z_t, z_{t-1}, a_{t-1}, o_t) \log \frac{p(z_t|z_{t-1}, a_{t-1}, o_t)}{q_\chi(z_t|z_{t-1}, a_{t-1})} \\
&= -D_{\text{KL}}(p(z_t|z_{t-1}, a_{t-1}, o_t)||q_\chi(z_t|z_{t-1}, a_{t-1}))
\end{aligned}
$$

We can obtain a variational lower bound on the empowerment

$$
\begin{aligned}
I(a_t; z_{t+k}|z_t) &= \int p(a_t, z_{t+k}, z_t) \log \frac{p(a_t|z_{t+k}, z_t)}{p(a_t|z_t)} \\
&= \int p(a_t, z_{t+k}, z_t) \left( \log \frac{q(a_t|z_{t+k}, z_t)}{p(a_t|z_t)} + \log \frac{p(a_t|z_{t+k}, z_t)}{q(a_t|z_{t+k}, z_t)} \right) \\
&\geq \int p(a_t, z_{t+k}, z_t) \log \frac{q(a_t|z_{t+k}, z_t)}{p(a_t|z_t)} \\
&= \int p(a_t, z_{t+k}, z_t) \log q(a_t|z_{t+k}, z_t) - \int p(z_t) p(z_{t+k}|a_t, z_t) p(a_t|z_t) \log p(a_t|z_t) \\
&= \mathbb{E}_{p(a_t, z_{t+k}, z_t)}[\log q(a_t|z_{t+k}, z_t)] + \mathbb{E}_{p(z_t)p(z_{t+k}|a_t, z_t)}[\mathcal{H}(p(a_t|z_t))]
\end{aligned}
$$

Here, $q(a_t|z_{t+k}, z_t)$ is the inverse dynamics model, and $\mathcal{H}(p(a_t|z_t))$ is the entropy of the policy.

We can obtain a variational lower bound on the reward model as

$$
\begin{aligned}
I(r_t; z_t) &= \int p(r_t, z_t) \log \frac{p(r_t|z_t)}{p(r_t)} \\
&= \int p(r_t, z_t) \left( \log \frac{q(r_t|z_t)}{p(r_t)} + \log \frac{p(r_t|z_t)}{q(r_t|z_t)} \right) \\
&= \int p(r_t, z_t) \log \frac{q(r_t|z_t)}{p(r_t)} + KL[p(r_t, z_t)||q(r_t, z_t)] \\
&\geq \int p(r_t, z_t) \log \frac{q(r_t|z_t)}{p(r_t)} \\
&= \int p(r_t, z_t) \log q(r_t|z_t) - \int p(r_t, z_t) \log p(r_t) \\
&= \mathbb{E}_{p(r_t, z_t)}[\log q(r_t|z_t)] + Constant
\end{aligned}
$$

Here, $q(r_t|z_t)$ is the reward decoder. Since the reward is a scalar, reconstructing it is computationally simpler compared to reconstructing high dimensional observations as in the BA bound for the observation model.

## A.4 Value Difference Result

We show that the difference in value function of the underlying MDP $Q^\pi(o, a)$ and the latent MDP $\hat{Q}^\pi(z, a)$, where $z \sim q_\phi(z|o)$ is bounded, under some regularity assumptions. Similar to the assumption in (Gelada et al., 2019), let the policy $\pi$ have bounded semi-norm value functions under the total variation distance $D_{\text{TV}}$ i.e. $|\hat{V}^\pi(z)|_{D_{\text{TV}}} \leq K$ and $|\mathbb{E}_{z_1 \sim P} V(z_1) - \mathbb{E}_{z_2 \sim Q} V(z_2)| \leq K D_{\text{TV}}(P, Q)$.

Define the forward dynamics learning objective as $\min_\chi L_T = \min_\chi D_{\text{KL}}(p(o'|o, a)||q_\chi(z'|z, a))$ and the reward model objective as $\min_\eta L_R = \min_\eta D_{\text{KL}}(p(r|o)||q_\eta(r|z))$

**Theorem 3.** *The difference between the Q-function in the latent MDP and that in the original MDP is bounded by the loss in reward predictor and forward dynamics model.*

$$\mathbb{E}_{(o,a)\sim d^\pi(o,a), z\sim q_\phi(z|o)}[Q^\pi(o,a) - \hat{Q}^\pi(z,a)] \leq \frac{\sqrt{L_R} + \gamma K\sqrt{L_T}}{1-\gamma}$$

*Proof.*

$$\mathbb{E}_{(o,a)\sim d^\pi(o,a), z\sim q_\phi(z|o)}[Q^\pi(o,a) - \hat{Q}^\pi(z,a)]$$

$$\leq \mathbb{E}_{(o,a)\sim d^\pi(o,a), z\sim q_\phi(z|o)}(|r(o,a) - \hat{r}(z,a)| + \gamma|\mathbb{E}_{o'\sim p(o'|o,a)}V(o') - \mathbb{E}_{z'\sim q_\chi(z'|z,a)}\hat{V}(z')|)$$

$$\leq \sqrt{D_{\text{KL}}(p(r|o)||q_\eta(r|z))} + \mathbb{E}_{(o,a)\sim d^\pi(o,a), z\sim q_\phi(z|o)}(\gamma|\mathbb{E}_{o'\sim p(o'|o,a), z'\sim q_\phi(z'|o')}[V(o') - \hat{V}(z')]|$$

$$+ \gamma|\mathbb{E}_{o'\sim p(o'|o,a), z'\sim q_\chi(z'|z,a)}[V(o') - \hat{V}(z')]|)$$

$$\leq \sqrt{L_R} + \mathbb{E}_{(o,a)\sim d^\pi(o,a), z\sim q_\phi(z|o)}(\gamma|\mathbb{E}_{o'\sim p(o'|o,a), z'\sim q_\phi(z'|o')}[V(o') - \hat{V}(z')]|)$$

$$+ \mathbb{E}_{(o,a)\sim d^\pi(o,a), z\sim q_\phi(z|o)}(\gamma K D_{\text{TV}}(p(o'|o,a)||q_\chi(z'|z,a)))$$

$$\leq \sqrt{L_R} + \gamma\mathbb{E}_{(o,a)\sim d^\pi(o,a), z\sim q_\phi(z|o)}\mathbb{E}_{o'\sim p(o'|o,a), z'\sim q_\phi(z'|o')}[V(o') - \hat{V}(z')]$$

$$+ \gamma K\sqrt{D_{\text{KL}}(p(o'|o,a)||q_\chi(z'|z,a))}$$

$$\leq \sqrt{L_R} + \gamma\mathbb{E}_{(o,a)\sim d^\pi(o,a), z\sim q_\phi(z|o)}[V(o) - \hat{V}(z)]) + \gamma K\sqrt{L_T}$$

$$\leq \sqrt{L_R} + \gamma\mathbb{E}_{(o,a)\sim d^\pi(o,a), z\sim q_\phi(z|o)}[Q(o,a) - \hat{Q}(z,a)]) + \gamma K\sqrt{L_T}$$

So, we have shown that

$$\mathbb{E}_{(o,a)\sim d^\pi(o,a), z\sim q_\phi(z|o)}[Q^\pi(o,a) - \hat{Q}^\pi(z,a)] \leq \frac{\sqrt{L_R} + \gamma K\sqrt{L_T}}{1-\gamma}$$

$\square$

We can see that, as $L_R, L_T \to 0$, the Q-function in the representation space of the MDP, $\hat{Q}$, becomes increasingly closer to that of the original MDP, $Q$. Since this result holds for all Q-functions, it also holds for the optimal $Q^*$ and $\hat{Q}^*$. This result extends sufficiency results in prior works Rakelly et al. (2021) to a setting with stochastic encoders and KL-divergence losses on forward dynamics and reward, as opposed to Wasserstein metrics (Gelada et al., 2019), and bisimulation metrics (Zhang et al., 2020).

## A.5 BEHAVIORAL SIMILARITY METRIC DETAILS

Given the sets $\mathcal{S}_z = \{z^i\}_{i=1}^n$ and $\mathcal{S}_{z_{\text{gt}}} = \{z_{\text{gt}}^i\}_{i=1}^n$, we construct complete graphs by connecting every vertex with every other vertex through an edge. The weight of an edge is the Euclidean distance between the respective pairs of vertices. The label of each vertex is an unique integer and corresponding vertices in the two graphs are labelled similarly. Let the resulting graphs be denoted as $G_z = (V_z, E_z)$ and $G_{z_{\text{gt}}} = (V_{z_{\text{gt}}}, E_{z_{\text{gt}}})$ respectively. Note that both these graphs are *shortest path* graphs, by construction.

Since the Euclidean distances in both the graphs cannot be directly compared, we scale the distances such that the shortest edge weight among all edge weights are equal in both the graphs. Scaling every edge weight by the same number doesn't change the structure of the graph. Now, we define the shortest path kernel to measure similarity between the two graphs, as follows

$$k(G_z, G_{z_{\text{gt}}}) = \frac{1}{|E_z|} \sum_{e_i \in E_z} \sum_{e_j \in E_{z_{\text{gt}}}} \hat{k}(e_i, e_j)$$

Let, $e_i = \{u_i, v_i\}$ and $e_j = \{u_j, v_j\}$. Here, the kernel $\hat{k}$ measures similarity between the edges $e_i, e_j$ and the labels on respective vertices. Let the function $l(\cdot)$ denote the labels of vertices and the weights of edges.

$$\hat{k}(e_i, e_j) = k_v(l(u_i), l(u_j))k_e(l(e_i), l(e_j))k_v(l(v_i), l(v_j)) + k_v(l(u_i), l(v_j))k_e(l(e_i), l(e_j))k_v(l(v_i), l(u_j))$$

Here, $k_v$ is a Dirac delta kernel i.e. $k_e(x,y) = 1 - \delta(x,y)$ (Note that $\delta(x,y) = 1$ iff $x = y$, else $\delta(x,y) = 0$) and $k_e$ is a Brownian ridge kernel, $k_e(x,y) = \frac{1}{c}\max(0, c - |x - y|)$, where $c$ is a large number.

So, in summary, the value of $\hat{k}(e_i, e_j)$ is low when the edge lengths between corresponding pairs of vertices in both the graphs are close. Hence, the value of $k(G_z, G_{z_{\text{gt}}})$ is low when a large number of pairs of corresponding vertices in both the graphs have the same edge length.

We expect methods that recover latent state representations which better reflect the true underlying simulator state (i.e., the positions of the joints) to have higher values according to this metric.

### A.6 DESCRIPTION OF THE DISTRACTORS

**ILSVRC dataset** (Russakovsky et al., 2015) The distractors used for the main experiments are natural video backgrounds with RGB images from the ILSVRC dataset. We use 200 videos during training, and reserve 50 videos for testing. An illustration of this for different environments are shown in Fig. 8. These images are snapshots of what is received by the algorithm (64x64x3 images) and no information about proprioceptive features is provided. While natural videos that are unrelated to the task might be easy to ignore, realistic scenes might have other elements that resemble the controllable elements, but are not actually controllable (e.g., other cars in a driving scenario).

To emulate this, we also add distractors that resemble other potentially controllable robots, as shown for example in Fig. 8 (top left and bottom right frames), but are not actually influenced by the actions of the algorithm. These are particularly challenging because the background video has a pre-recorded motion of the same (or similar) agent that is being controlled. We include such challenging agent-behind-agent background videos with a frequency of 1 in very 3 videos.

For the experiments in Table 1, examples of different levels of distractors are shown in Fig. 9. The three different levels have distractor windows of size 32x32, 40x40, and 64x64 respectively.

**Kinetics dataset** (Kay et al., 2017) For comparing directly with results in prior works DBC (Zhang et al., 2020), and TIA (Fu et al., 2021), we evaluate on the setting where the random videos are grayscale images from the Kinetics dataset *driving car* class. The settings are same as in the prior works and we compare directly to the results in the respective papers.

### A.7 TRAINING AND NETWORK DETAILS

The agent observations have shape 64 x 64 x 3, action dimensions range from 1 to 12, rewards per time-step are scalars between 0 and 1. All episodes have randomized initial states, and the initial distractor video background is chosen randomly.

We implement our approach with TensorFlow 2 and use a single Nvidia V100 GPU and 10 CPU cores for each training run. The training time for 1e6 environment steps is 4 hours on average. In comparison, the average training time for 1e6 steps for Dreamer is 3 hours, for CDreamer is 4 hours, for TIA is 4 hours, for DBC is 4.5 hours, and for DeepMDP is 5 hours.

The encoder consists of 4 convolutional layers with kernel size 4 and channel numbers 32, 65, 128, 256. The forward model consists of deterministic and stochastic parts, as in standard in visual model-based RL (Hafner et al., 2019b;a). The stochastic states have size 30 and deterministic state has size 200. The compatibility function $f_\theta(z_t, o_{t'}) = \exp(z_t^T W_\theta z_{t'})$ for contrastive learning is learned with a 200 x 200 $W_\theta$ matrix. Here, $z_{t'}$ is the encoding of $o_{t'}$. All other models are implemented as three dense layers of size 300 with ELU activations.

We use ADAM optimizer, with learning rate of 6e-4 for the latent-state space model, and 8e-5 for the value function and policy optimization. The hyper-parameter $c_0$ for the prioritized information constraint is set to 1000, after doing a grid-search over the range $[100, 10000]$ and observing similar performance for 1000 and 10000. 100 training steps are followed by 1 episode of interacting with the environment with the currently trained policy, for observing real transitions. The dataset is initialized with 7 episodes collected with random actions in the environment. We kept the hyper-parameters of `InfoPower` same across all environments and distractor types.

For fair comparisons, we kept all the common hyper-parameter values same as Dreamer (Hafner et al., 2019a). This was the protocol followed in prior works TIA (Fu et al., 2021) and Ma et al.

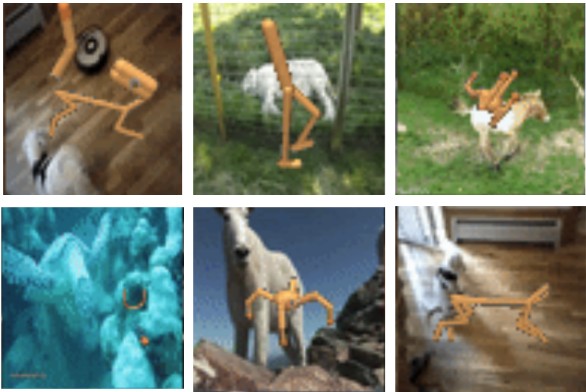

Figure 8: Illustration of the natural video background distractors used in our experiments. The videos change after every 50 time-steps. Some video backgrounds (for example the top left and the bottom right) have more complex distractors in the form of agent-behind-agent i.e. an agent of similar morphology moves in the background of the agent being controlled.

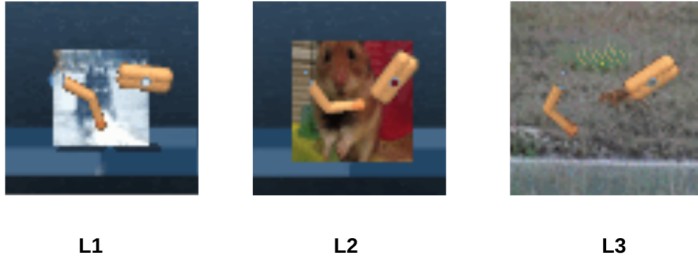

**L1**          **L2**          **L3**

Figure 9: Illustration of the different levels of distractors. L1, L2, and L3 respectively have distractor windows of size 32x32, 40x40, and 64x64.

(2020). For the baseline TIA, we tuned the environment-specific parameters $\lambda_{R_{adv}}$ and $\lambda_{O_s}$ as mentioned in Table 2 of (Fu et al., 2021). For $\lambda_{R_{adv}}$, we performed a gridsearch over the range $10k - 50k$ and for $\lambda_{O_s}$ we performed a gridsearch over the range $1 - 3$ to obtain the parameters for best performance, which we used for the plots in Fig. 4 and Fig. 11. Respectively for walker-walk, cheetah-run, finger-spin, quadruped-walk, hopper-stand, ball-in-a-cup-catch, cartpole-balance-sparse, quadruped-run, the values of $\lambda_{R_{adv}}$ are 30k,30k,20k,40k,30k,40k,20k,30k. The values of $\lambda_{O_s}$ are 2,2,1.5,2.5,2.5,2,2,2.

For baseline DBC (Zhang et al., 2020), we kept all the parameters same as in Table 2 of the paper, because DBC does not have any environment-specific parameters and the same values were used for all environments in the DBC paper. The DeepMDP agent and its hyperparameters are adapted from the implementation provided in the github repo of DBC (Zhang et al., 2020).

### A.8 CONTRASTIVE LEARNING

Since the distracting backgrounds are changing videos of a certain duration, for contrastive learning, we contrast both across time and across batches. Contrasting across time helps model invariance against temporally contiguous distractors (for example in the same video) while contrasting across batches helps in learning invariance across different videos. Concretely, we sample batches from the dataset $^{(i)}\{(a_t, o_t, r_t)\}_{t=1}^{H}$, where $i$ denotes a batch index, and for each observation $o_t$ in batch $i$, all observations $o_{t'}$ both in batch $i$ and in other batches $j \neq i$ are considered negative samples.

### A.9 RESULT ON VARYING DISTRACTOR LEVELS

We consider different levels of distractors by varying the size of the window where distractors in the background are active. Fig. 9 in the Appendix illustrates this visually. Table 1 shows results for

Table 1: DM Control Walker Stand with natural video distractors at different levels. We tabulate the rewards (Rew) and behavioral similarity (Sim) at 500k and 1M steps of training. Higher is better for both Rew and Sim.

| Name | Levels | Rew@500k | Rew@1M | Sim@500k | Sim@1M |
|---|---|---|---|---|---|
| Dreamer | L1 | $197 \pm 31$ | $240 \pm 27$ | $0.73 \pm 0.02$ | $0.74 \pm 0.01$ |
| DBC | L1 | $261 \pm 25$ | $390 \pm 34$ | $0.75 \pm 0.03$ | $0.74 \pm 0.02$ |
| C-Dreamer | L1 | $291 \pm 34$ | $590 \pm 26$ | $0.73 \pm 0.02$ | $0.79 \pm 0.01$ |
| DeepMDP | L1 | $263 \pm 31$ | $340 \pm 22$ | $0.74 \pm 0.02$ | $0.75 \pm 0.03$ |
| InfoPower | L1 | $\mathbf{397 \pm 22}$ | $\mathbf{650 \pm 100}$ | $\mathbf{0.82 \pm 0.01}$ | $\mathbf{0.84 \pm 0.03}$ |
| Dreamer | L2 | $180 \pm 36$ | $197 \pm 20$ | $0.56 \pm 0.03$ | $0.59 \pm 0.02$ |
| DBC | L2 | $221 \pm 28$ | $320 \pm 33$ | $0.65 \pm 0.02$ | $0.66 \pm 0.02$ |
| C-Dreamer | L2 | $282 \pm 30$ | $550 \pm 66$ | $0.63 \pm 0.01$ | $0.71 \pm 0.02$ |
| DeepMDP | L2 | $213 \pm 34$ | $300 \pm 25$ | $0.59 \pm 0.02$ | $0.61 \pm 0.04$ |
| InfoPower | L2 | $\mathbf{394 \pm 30}$ | $\mathbf{644 \pm 101}$ | $\mathbf{0.77 \pm 0.03}$ | $\mathbf{0.77 \pm 0.03}$ |
| Dreamer | L3 | $140 \pm 52$ | $157 \pm 10$ | $0.32 \pm 0.02$ | $0.33 \pm 0.03$ |
| DBC | L3 | $165 \pm 20$ | $221 \pm 24$ | $0.45 \pm 0.01$ | $0.48 \pm 0.01$ |
| C-Dreamer | L3 | $231 \pm 39$ | $485 \pm 86$ | $0.58 \pm 0.02$ | $0.64 \pm 0.01$ |
| DeepMDP | L3 | $153 \pm 23$ | $212 \pm 28$ | $0.44 \pm 0.02$ | $0.49 \pm 0.01$ |
| InfoPower | L3 | $\mathbf{389 \pm 20}$ | $\mathbf{624 \pm 80}$ | $\mathbf{0.71 \pm 0.01}$ | $\mathbf{0.74 \pm 0.03}$ |

Table 2: Comparison of baselines TIA and DBC with InfoPower on environments with video distractors from the Kinetics dataset *Driving Car* class. The results for TIA and DBC are from the respective papers.

| | TIA | | DBC | | InfoPower | |
|---|---|---|---|---|---|---|
| | 500k | 800k | 500k | 800k | 500k | 800k |
| Walker-Run | 389±45 | 602±40 | 165±92 | 210±32 | **417±22** | **630±33** |
| Cheetah-Run | 384±149 | **579±172** | 261±64 | 277±81 | **425±97** | 572±158 |
| Hopper-Stand | 338±221 | 581±214 | 110±92 | 207±120 | **395±140** | **615±176** |
| Ball-in-a-Cup | 201±197 | 115±110 | – | – | **255±104** | **263±116** |
| Walker-Walk | 878±37 | 948±28 | 545±57 | 630±68 | **925±18** | **972±26** |
| Finger-Spin | 371±96 | 487±107 | **801±10** | **832±4** | 795±34 | 787±58 |
| Hopper-Hop | 47±44 | **72±68** | 5±12 | 0±0 | **85±26** | **77±74** |

the different approaches on varying distractor levels. The size of the frame for distractors at levels L1, L2, and L3 respectively are 32x32, 40x40, and 64x64. We observe that the baselines perform worse as the distractor window size increases. While, for InfoPower, the performance decrease with increasing level is minimal. This indicates the effectiveness of InfoPower in filtering out background distractors from the observations while learning latent representations. Note that the distractors in Figs. 4 and 6 are of level 3 i.e. same size as the observations 64x64.

## A.10 RESULTS ON DISTRACTORS FROM THE KINETICS DATASET

In this section, we evaluate InfoPower on the same distractors used in prior works, DBC and TIA. The distractor videos are from the Kinetics dataset (Kay et al., 2017) *Driving Car* class, and converted to grayscale. From Table 2 we see that InfoPower slightly outperforms the baselines. We believe the performance gap is not as significant as Fig. 4 because the distractors being grayscale images, and not RGB might be easier to ignore by the baselines as well as InfoPower.

## A.11 SETTING WITH ONLY SHIFTED AGENT BEHIND AGENT DISTRACTORS

In this section we evaluate InfoPower with the baselines only for the challenging setting of agent-behind-agent distrators where the background contains the same agent being controlled (a walker or a cheetah). An illustration of this is shown in Fig. 10. From the results in Table 3, we see that InfoPower significantly out-

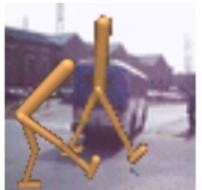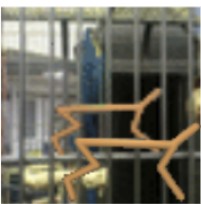

Figure 10: Ilustration of the distractor setting for results in Table 3.

performs the competitive baselines both at 500k and 1M environment interactions. This suggests the utility of `InfoPower` in separating out informative elements of the observation from uninformative ones even in challenging settings.

Table 3: Evaluation of `InfoPower` and the most competitive baselines in a suite of challenging distractors where the background contains a shifted version of the agent being controlled. Results are averaged over 4 random seeds. Fig. 10 shows a visualization of the distractors for this setting.

|  | Walker-Walk-shifted | | Cheetah-Run-shifted | |
|---|---|---|---|---|
|  | 500k | 1M | 500k | 1M |
| DBC | 35±15 | 104±28 | 41±10 | 78±38 |
| CDreamer | 108±36 | 164±45 | 73±44 | 110±52 |
| TIA | 79±42 | 152±33 | 65±24 | 115±28 |
| InfoPower | **168±48** | **278±42** | **130±46** | **207±35** |

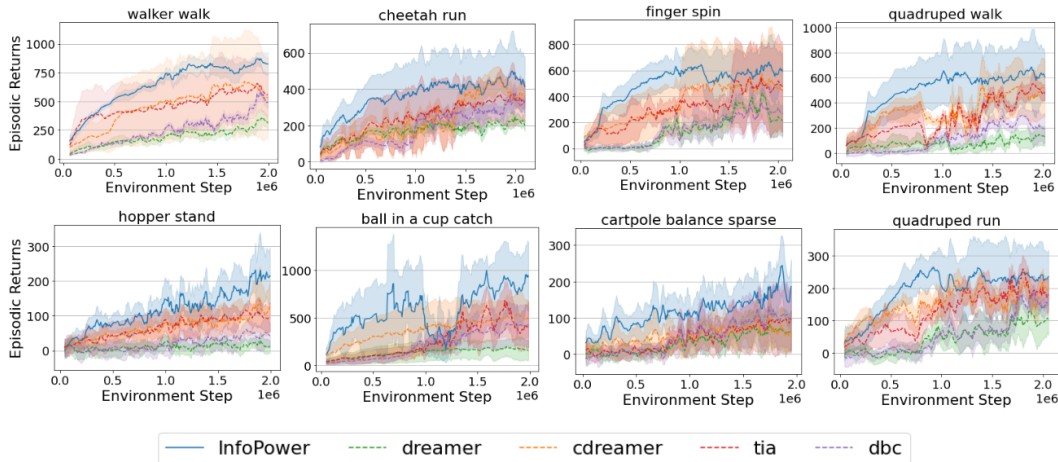

Figure 11: Evaluation of `InfoPower` and baselines with empowerment in policy learning. The setting corresponds to DeepMind Control tasks with RGB natural video distractors in the background (same as that in Figure 5 of the main paper). The x-axis denotes the number of environment interactions and the y-axis shows the episodic returns.

## A.12 COMPARISON OF BASELINES WITH EMPOWERMENT IN POLICY LEARNING

In this section, we compare against modified versions of the baselines, such that we include empowerment in the policy learning of the baselines. This is the only modification we make to the baselines, and keep everything else unchanged. We described in section 3.3 that inclusion of empowerment in the objective for visual model-based RL (by modifying both the representation learning objective and the policy learning objective) is an important contribution of our paper. The aim of this experiment is to disentangle the benefits of empowerment in representation learning, for `InfoPower` with that in policy learning.

We have plotted the results in Fig. 11. We can see that the performance of the baselines is slightly better than Fig. 4 because of the inclusion of empowerment in the policy learning objective which as we motivated previously in section 3.4 helps with exploring controllable regions of the state-space. Note that by adding empowerment, we have effectively modified the baselines.

## A.13 COMPARISON ON ENVIRONMENTS WITHOUT DISTRACTORS

We compare the performance of `InfoPower` with a state-of-the-art visual model-based RL method, Dreamer in the default DM Control environments without distractors. We see from Table 4 that `InfoPower` performs slightly better or similar to Dreamer in all the environments consistently. This result complements our main paper result on environments with distractors and confirms that the benefit of `InfoPower` in challenging distracting environments does not come at a cost of performance in simpler non-distracting environments.

Table 4: Comparison of `InfoPower` with state-of-the-art visual model-based RL method, Dreamer in the default DM Control environments without distractors. We tabulate reward at 500k and 1M environment interactions. Results are averaged over 4 random seeds. Higher is better.

|  | Dreamer | | InfoPower | |
|---|---|---|---|---|
|  | Rew @ 500k | Rew @ 1M | Rew @ 500k | 1M |
| Walker Walk | 890±52 | **985± 10** | **910±41** | 980± 17 |
| Cheetah Run | 580±190 | **811±95** | **586±142** | 810±102 |
| Finger Spin | 570±187 | 573±96 | **590±172** | **598±85** |
| Quadruped Walk | 592±45 | 646±50 | **623±65** | **666±47** |
| Hopper Stand | 701±68 | 906±33 | **715±60** | **930±39** |
| Ball in a Cup | 871±102 | 910±48 | **913±67** | **939±52** |
| Cart-Pole Sparse | 742±58 | 849±88 | **776±40** | **866±80** |
| Quadruped Run | 450±47 | **515±65** | **468±42** | 510±61 |

## A.14 Additional ablations

In this section, we consider additional ablations, namely a version of of `InfoPower` that does not have the empowerment objective, a version of `InfoPower` that does not have the MI maximization objective. We see from Table 5 that removing either of these terms drops performance.

Table 5: DM Walker Stand with natural video distractors

| Name | Obs bound | Rew@500k | Rew@1M | Sim@500k | Sim@1M |
|---|---|---|---|---|---|
| Dreamer | BA | $140 \pm 52$ | $157 \pm 10$ | $0.32\pm0.02$ | $0.33\pm0.03$ |
| Contrastive-Dreamer | NCE | $231 \pm 39$ | $485 \pm 86$ | $0.58\pm0.02$ | $0.64\pm0.01$ |
| NonGenerative-Dreamer | NWJ | $280 \pm 25$ | $480 \pm 93$ | $0.59\pm0.01$ | $0.62\pm0.05$ |
| InfoPower | NCE | $254 \pm 27$ | $481 \pm 30$ | $0.69\pm0.03$ | $0.72\pm0.02$ |
| InfoPower | NWJ | $\mathbf{389 \pm 20}$ | $\mathbf{624 \pm 80}$ | $0.71\pm0.01$ | $0.74\pm0.03$ |
| Inverse only | None | $282 \pm 70$ | $367 \pm 24$ | $0.63\pm0.04$ | $0.66\pm0.03$ |

