# OpenReview forum: "Information Prioritization through Empowerment in Visual Model-based RL"
_ICLR.cc/2022/Conference — ICLR 2022 Poster_

### Official Review · Reviewer_EU46 · 2021-10-31

**Correctness:** 4
**Technical Novelty And Significance:** 3
**Empirical Novelty And Significance:** 3
**Recommendation:** 6
**Confidence:** 4

**Main Review:**

#### **Strengths**
- The paper tackles a very important problem in the topic of learning to control from visual observations. In real-life scenarios, the observations are likely to contain distractors or task-irrelevant information, and being able to focus on relevant aspects of the environment in those cases is crucial.
- The proposed use of information empowerment for learning representations and exploration is novel to the best of my knowledge.
- The experimental results look promising.

#### **Weaknesses**
- The first major issue of the paper is the lack of details in the presentation of the practical algorithm. Specifically, it is not clear from the text what components/networks constitute the InfoPower model. In Section 3.4, the authors derived lower bounds of MI terms that will be used to optimize the model. However, it is very confusing to read because the authors have not specified what $p$, $q_{\mathcal{X}}$, $q_{\eta}$, and $q_{\psi}$ are and how they are parameterized. Moreover, these lower bounds also need to be derived in more detail as they are not straightforward from the text. In Appendix A.8, the authors presented the training and network details but again it does not provide enough information to fully understand the components of InfoPower.
- The second major issue lies in the overall objective function, as there are multiple terms being optimized together. First of all, why do we need to maximize $I(O_t; Z_t)$ if we are already minimizing $I(i_t; Z_t \mid Z_{t-1}, A_{t-1})$, which encourages encoding the predictable parts of the observations? In previous works [1,2,3,4] people have found that maximizing $I(Z_t; Z_{t-1}, A_{t-1})$, which is similar to minimizing $I(i_t; Z_t \mid Z_{t-1}, A_{t-1})$, allows to learn a much better representation compared to maximizing $I(O_t; Z_t)$. While being a contrastive loss, $I(O_t; Z_t)$ does not behave differently from a reconstructions loss, as it requires the encoder to retain as much information in the observation as possible, which is contradictory to the motivation of the paper.
- The use of both $I(i_t; Z_t \mid Z_{t-1}, A_{t-1})$ and $I(A_{t-1}; Z_{t} \mid Z_{t-1})$ also seems redundant. In fact, $I(A_{t-1}; Z_{t} \mid Z_{t-1}) = I(A_{t-1}, Z_{t-1}; Z_t) - I(Z_t; Z_{t-1})$ and therefore maximizing $I(A_{t-1}; Z_{t} \mid Z_{t-1})$ will also maximize $I(A_{t-1}, Z_{t-1}; Z_t)$, which has similar effects to minimizing $I(i_t; Z_t \mid Z_{t-1}, A_{t-1})$.

#### **Other comments and suggestions**
- While the assumption that the features of the observations influenceable by the agent through its actions are important for control is true in the experiments that the paper considers, it is not true in general. Consider a robot trying to navigate from A to B by using visual information, its actions such as turning left or right can actually influence the distractors (trees, irrelevant subjects, etc.) in the scene that it observes. However, this can be left for future work.
- The paper is missing some important references. PC3 [2] is one of the first methods that proposed to use mutual information/contrastive learning to learn representations for control. TPC [1] is a very relevant paper that was also proposed to address the problem of learning representations for visual model-based RL with distractors in the observations. TPC outperforms Dreamer, CDreamer and DBC and should serve as a stronger baseline. PI-SAC [3] is also a method related to the line of work that uses mutual information to learn representations for RL. In [4] the authors theoretically investigated different mutual information objectives used to learn representations for RL.
- I want to see the performance of InfoPower vs the baselines in the standard setting (no natural backgrounds), as a good method should work well in both standard and natural background settings.

[1] Nguyen, T., Shu, R., Pham, T., Bui, H. and Ermon, S., 2021. Temporal Predictive Coding For Model-Based Planning In Latent Space. arXiv preprint arXiv:2106.07156.

[2] Shu, R., Nguyen, T., Chow, Y., Pham, T., Than, K., Ghavamzadeh, M., Ermon, S. and Bui, H., 2020, November. Predictive coding for locally-linear control. In International Conference on Machine Learning (pp. 8862-8871). PMLR.

[3] Lee, K. H., Fischer, I., Liu, A., Guo, Y., Lee, H., Canny, J., & Guadarrama, S. (2020). Predictive information accelerates learning in rl. arXiv preprint arXiv:2007.12401.

[4] Rakelly, K., Gupta, A., Florensa, C. and Levine, S., 2021. Which Mutual-Information Representation Learning Objectives are Sufficient for Control?. arXiv preprint arXiv:2106.07278.


**Summary Of The Paper:**

The paper proposes a new non-reconstruction method for model-based RL from high-dimensional observations. The main idea is to make use of an information empowerment objective that prioritizes encoding parts of the environment that are influenceable by the actions. This allows the model to focus on functionally relevant information and filter out distractors. The same empowerment term can also be used to promote faster exploration when the reward signal is sparse. The proposed method outperformed the existing baselines in difficult Deepmind control tasks with natural video backgrounds.

**Summary Of The Review:**

In general, the proposed method is promising, but the current presentation of the paper has major issues that need to be addressed and clarified.

---

> ### Author Response · Authors · 2021-11-16
> **Author response**
>
> We thank the reviewer for their comments and suggestions. We have updated the paper based on the comments, with major updated parts highlighted in blue. In particular, we have clarified the definition of each model in section 3.4 of the paper, and clearly referenced appendix A.7 where additional details about the specific architectures of the models are present. Please find below our detailed responses to the reviewer’s comments
>
> - **Details about the practical algorithm** We have now clarified the definition of each model in section 3.4 of the paper. Here, $q_\phi(z_t|o_t)$ is the observation encoder, $ q_\psi(a_{t-1}| z_{t},z_{t-1})$ is the inverse dynamics model, $q_\eta(r_t|z_t)$ is the reward decoder, and $q_\chi(z_t|z_{t-1},a_{t-1})$ is the forward dynamics model.  The inverse model helps in learning representations such that the chosen action is predictable from successive latent states. The forward dynamics model helps in predicting the next latent state given the current latent state and the current action, without having the next observation. The reward decoder predicts the reward (a scalar) at a time-step given the corresponding latent state. We use dense networks for all the models, with the architectures described in section A.7 of the Appendix. We indeed provide derivations for the lower bounds in section A.3 of the Appendix - we would like to draw the reviewer’s attention to this section in case they missed it.
>
> - **Regarding the $I(\mathcal{O}_t;Z_t)$ term** Thank you for the references [1-4]. We were aware of [4] and had cited it in our paper - we have now updated the related works to cover papers [1-3] as well which are very relevant. In our experiments, we found that removing the  $I(O_t;Z_t)$ term drops performance (the last row of the table in section A.13). We believe it is important to have this term in order to learn the encoder mapping from observations to latents in the early stages of training when the reward signal and forward predictive model (through $- I(i_t;Z_t|Z_{t-1}, A_{t-1} $) are not accurate enough. This observation is was also made in prior works based on Dreamer [a,b] where there is a specific objective based on $I(O_t;Z_t)$ (either through reconstruction or contrastive learning) for learning the encoder.
>
> - We agree with the reviewer that the term $I(O_t;Z_t)$ encourages encoding as much from the observations as possible, however, kindly note that we have a constrained optimization problem, where we are explicitly encouraging non-distractors to be encoded (in section 3.3). This ensures that the term $I(\mathcal{O}_t;Z_t)$ only improves representation learning for the encoder while *prioritizing* the encoding of non-distractors.
>
> - **Regarding $- I(i_t;Z_t|Z_{t-1}, A_{t-1})$ and $I(A_{t-1};Z_{t}|Z_{t-1})$** We would like to kindly point out that these two terms are not equivalent - maximizing one does not necessarily maximize the other. In the equation the reviewer mentioned $I(A_{t-1};Z_{t}|Z_{t-1}) = I(A_{t-1},Z_{t-1};Z_{t}) - I(Z_t;Z_{t-1})$, kindly note that maximizing the RHS does not automatically maximize $I(A_{t-1},Z_{t-1};Z_{t})$ because the RHS can be maximized even when $I(A_{t-1},Z_{t-1};Z_{t})$ is held constant and $I(Z_t;Z_{t-1})$ is minimized. Intuitively, the term $\max_Z I(A_{t-1},Z_{t-1};Z_{t})$ is maximizing predictability of the next latent state given the current latent state and the current action, whereas, the term $\max_Z I(A_{t-1};Z_{t}|Z_{t-1})$ is maximizing the predictability of the chosen action given current and past latent states. This distinction between the two objectives has also been described in prior work [4]. Also, kindly note that we use the term $I(A_{t-1};Z_{t}|Z_{t-1})$ for planning as well via $\max_A I(A_{t-1};Z_{t}|Z_{t-1})$ described in section 3.3 and 3.4. Integrating this empowerment objective in visual model-based RL is a key contribution of our paper.
>
> - Thank you for suggesting the related papers in [1-3]. We have now cited and discussed them in the related works section.  The TPC paper is very relevant but we were unable to do an empirical comparison with this paper during the author response period, as the code is not publicly available, and it would take us some time to re-implement it. We will definitely do this, and incorporate the results in the revised paper.
>
> - Kindly note that we had already included the comparisons of InfoPower in the standard setting (without distractors) in our paper. This is in Appendix A.12 Table 4. We compare InfoPower with the reference Dreamer as it is the most competitive visual model-based RL baseline in the setting without distractors.
>
> Kindly let us know if there is anything else that we can better clarify for the reviewer, for a revised assessment of our paper.
>
> [a] Ma, X., Chen, S., Hsu, D., & Lee, W. S. Contrastive variational model-based reinforcement learning for complex observations. CoRL 2020
>
> [b] Fu, Xiang, et al. "Learning task informed abstractions." ICML 2021

---

> > ### Comment · Reviewer_EU46 · 2021-11-21
> > **Updated score**
> >
> > Thank you for the detailed response. After reading through all the bullet points, I decided to raise my score to Weak accept. The clarification issue has been resolved, and my concerns about the objective functions have been fairly addressed. I want to note that the official implementation of TPC is released here https://github.com/VinAIResearch/TPC-tensorflow and the authors should be able to compare with this baseline quickly. I have two additional questions:
> > 1) In the main text, the authors mentioned that InforPower uses both deterministic and stochastic paths, which is similar to Dreamer. So what part the authors are using to represent $z$?
> > 2) What is the final objective function used to learn the representation? Did the authors use any special weighting mechanism between all the mutual information terms? It seems to me that we need to pay careful attention when optimizing several MI objectives.

---

> > > ### Author Response · Authors · 2021-11-23
> > > **Thank you!**
> > >
> > > Thank you so much for your updated assessment. We are glad that the concerns have been addressed. Thank you for pointing us towards the implementation of TPC - we will use it for comparison in the paper. Please find our answers to the additional questions below:
> > >
> > > 1. In relation to Dreamer, the latent $z$ in our paper corresponds to $s$ in Dreamer, which represents the stochastic paths. We will make this more clear in the paper. Thank you for the question!
> > >
> > > 2. The final objective for representation learning is the combination of the lower bounds to terms in section 3.3 - we did not consider any special weighing mechanism as that would introduce a lot more hyper-parameters requiring a much larger compute budget to tune. We definitely agree that it might be possible to improve performance further by considering special weightings and carefully optimizing for the weight hyper-parameters. We would like to explore this further in the updated paper. Thank you for the suggestion.

---

> ### Author Response · Authors · 2021-11-19
> **Request for discussion**
>
> Dear reviewer,
>
> Since the author response period is closing very soon, we wanted to send a gentle request for discussion. Please find our responses to the original review below, and the revised paper that we have uploaded. It will be very helpful if you could let us know whether we have addressed your concerns and if we can provide any additional clarification for a revised assessment of our paper.
>
> Thanks,
> Authors

---

### Official Review · Reviewer_sJgS · 2021-11-01

**Correctness:** 4
**Technical Novelty And Significance:** 3
**Empirical Novelty And Significance:** Not applicable
**Recommendation:** 8
**Confidence:** 3

**Main Review:**

Strengths:

- I like the overall idea. It's not too complicated and the results seem very promising.
- With one exception, the illustrations and figures are helpful.

Weaknesses:

- W1. The paper is structured badly. It feels like the whole paper was written, it came out as 11 pages and hours before the deadline, hastily important sections were moved to the appendix. For example there are 2 theorems in the main paper but no proof. Why format them as "theorems" at all in the main body of the paper if there's no proof right there? I'd just trim this down, only include the description of these in prose and move the "theorem" and proof both into the appendix. I also felt like the theory part in the main body of the paper was a bit shallow because it's long and verbose but doesn't feel complete and as easy to follow as I would have expected based on how much space it takes up. Another example of this is section 4.2 where you "introduce" a new similarity metric but never explain it in the main body of the paper. Figure 4 doesn't make any sense without consulting the appendix. This needs to go back into the main body of the paper!
- W2. The writing could have definitely used a proofreading. There are tons of errors in the main doc and a lot more in the appendix. Also some expressions are a bit handwavy and could have used a bit more rigorous writing. E.g. in the intro, you wrote "... modeling the observation space so as to most quickly capture the functionality relevant factors... " - (a) You usually don't model the observation space but "states" or a "latent representation" that's inferred from the observations; (b) "quickly"? Why quickly? That's not really explained; (c) "functionality relevant factors" - you probably mean "functionally-relevant factors", right? (d), functionally relevant to what? To a task, to control, to an understanding of the robot's odometry? You have a good example of this later in Figure 2 but here, without any context, this is too imprecise language.
- W3. Fundamentally, I think to show your assumption properly, you should've included a task where the observations consists quantifiably of those 3 different factors: S+: images of the robot itself that can be affected by actions, S-: elements that can't be controlled by the agent but influences the reward, DS-: distractors that don't affect the reward. In your current setup, there's only S+ and DS- because there is a random video playing in the background. How about this: Include obstacles on the ground (e.g. in front of halfcheetah and walker) that obviously cannot be affected by actions but would make the agent stumble if ignored. Or alternatively or additionally, instead of a random video, include a video of the agent itself but shifted up or down. These would be a harder but more informative test if your method can actually learn to separate out informative state elements from uninformative ones.
- W4. Why did you include a model-based version of DBC and not DBC itself? If you compare the results from Fig.3 in the DBC paper with your results in Fig. 5, I can see that DBC is supposed to be significantly better than this. The whole purpose of DBC is to remove distractions from the observations when learning a policy and it is already very sample-efficient (it can reach the performance of your model in 1e5 steps instead of 1e6 or 2e6). And speaking of which, in the appendix you mention training for 10e6 steps. Why didn't you include these results? What made you pick 1e6/2e6 for comparison and not 1e5 or 1e7? Feels a bit cherry-picky.

Nitpicks & Questions:

- Q1. Section 4.4 "the window size for distractors" is not explained.
- Q2. Spacing of Fig.3's caption needs to be fixed.
- Q3. The algorithm 1 block needs to be improved a lot. E.g. you initialize dataset D but not the policy and not lambda? Also what is lambda, why is it important there? Why is the environments not stepped until T but only until T-1? What is the line below "Compute latents" refer to? Update your model parameters? If that's the case then you need to specify that these are your model parameters. Where is the loss calculated? Computing the latents is not inherently also calculating the loss.

**Summary Of The Paper:**

The paper introduces a variation on the DREAMER model-based RL architecture that aims to remove elements from the latent state representation that cannot be affected by the actor. To realize this, there is an inverse model objective included in the value function.

**Summary Of The Review:**

UPDATE (2021-11-21): The authors have incorporated all of the suggested changes and addressed my concerns reasonably, which is why I'm now recommending acceptance.

OLD: The paper is overall okay. I don't think it's remotely in its best possible shape (in terms of structure, writing, Figure 4, etc.) but this is nothing that can't be solved with a couple hours editing and rearranging things. The main idea is good, nothing in the paper appears to me a major flaw and so I'm mildly recommending acceptance and strongly recommend polishing!

---

> ### Author Response · Authors · 2021-11-16
> **Author response**
>
> We thank the reviewer for their detailed feedback and suggestions. We have restructured some sections of the paper based on the suggestions, and updated the draft with typo corrections and new results. The major edited parts are highlighted in blue. Please find below our detailed responses and clarifications
>
> - **Updated the structuring of the paper** Thank you for the very helpful suggestions. We have now moved the theorems including the proofs to the Appendix, while forward-referencing and explaining the content of the theorems in the relevant sections of the main paper. We have removed Fig.4, and used the space to clearly describe the metric, by bringing in details from the Appendix to the main paper. We hope this structure is now more clear and self-contained.
>
> - **We have edited the typos in the text** Thanks for pointing out the parts in the Introduction that can be improved. We have now made major updates to the draft. In particular, the line pointed out by the reviewer now reads as *“While large neural network models have made progress on this problem [citations], sample-efficient learning necessitates some mechanism to prioritize modeling latent representations from observations such that functionally-relevant factors for the task can be captured”.*
>
> - **Experiment where the agent is shifted up/down** We have included results for this setting, where the background has the same agent (Walker/Cheetah) shifted in the frame. This is in Table 3 (section A.10) of the revised paper. The results on this setting show clear benefits of InfoPower over the baselines indicating better prioritization for capturing task relevant factors from the observations.
>
> - **Corrected typos in the text and numbers** We apologize for the typos in the original submission. The DBC baseline is indeed the original DBC algorithm and not a model-based modification - the phrase was present in a dated version of our paper, which unfortunately remained in the submitted paper. However, kindly note that the results in our original paper cannot be directly compared to that in the DBC paper because DBC used a set of grayscale distractors while we used RGB distractors. In order to facilitate a more direct comparison, we have now included results on the same distractor settings in Table 1 and section 4.4 of the paper. This setting was used in two recent baselines DBC and TIA, and we compare directly to their published results. Regarding the training time of 4 hours in the Appendix, it should be 1e6 steps and not 10e6 steps - we have corrected this. We only trained all the models for 2e6 steps and not 1e7 steps, so there was no cherry-picking for the reported results. We hope this clarifies the reviewer’s concern.
>
> - We have addressed spacing issues and typos in the text of the paper, and improved Algorithm 1 with details. In particular, Algorithm 1 now clearly lists that the model parameters and the dual variable $\lambda$ are initialized in the beginning. The line below “compute latents” refers to the model parameters, and we have referenced section 3.4 where we show how to compute the Lagrangian (loss function) for gradient descent.
>
> Kindly let us know if you have any suggestions about further improving the paper, and we would be very happy to incorporate them.

---

> > ### Comment · Reviewer_sJgS · 2021-11-22
> > **Updating my score (6->8)**
> >
> > Dear Authors, thanks for taking the time and addressing my concerns.
> > I'm changing my score to an 8 and I'm not strongly recommending acceptance.

---

> > > ### Author Response · Authors · 2021-11-23
> > > **Thank you!**
> > >
> > > Thank you so much for the updated score. We are glad that your concerns are addressed. Thank you for your time.

---

### Official Review · Reviewer_8ezz · 2021-11-03

**Correctness:** 4
**Technical Novelty And Significance:** 3
**Empirical Novelty And Significance:** 3
**Recommendation:** 8
**Confidence:** 3

**Main Review:**

I appreciate the idea of using the empowerment objective for both representation learning and policy learning. The agent-behind-agent distractors are also novel and more challenging than previous work. However, I have some concerns regarding the experiments and the clarity of the paper.
- While most of the baselines were tested in previous work using the same set of videos from the Kinetics dataset as background distractions, this paper uses a different set of videos without providing enough details. For example, how many videos are there? Are the videos obtained from some specific dataset? Do training and evaluation use the same set of videos? This makes it hard to compare with published results. In particular, some baselines require quite extensive hyperparameter tuning depending on the dataset, and TIA even requires separate hyperparameters for each domain in DMC. The paper did not include any detail about how the baselines were tuned. From the experiments, I do not see the need to change the background videos from previous work, so I think it would be better to compare with the baselines on the same dataset used in DBC and TIA. As for the agent-behind-agent distractors, I think it would be better to have a separate dataset focusing exclusively on this challenging setting, rather than mixing the challenging cases with standard ones.
- The comparison to baselines seems unfair, as the baselines do not use empowerment for policy learning. In fact, the ablation study in Figure 7 shows that removing the empowerment for policy learning significantly decreases performance of the proposed model, even on dense reward tasks like Walker Walk, where the ablated version underperforms C-Dreamer and TIA (compare to Figure 5). The paper can be strengthened by adding empowerment to the baselines.
- The paper claims to "outperform state-of-the-art model-based RL approaches by an average of 20% in terms of episodic returns at 1M environment interactions with 30% higher sample efficiency at 100k interactions". I do not see how these numbers are obtained.
- In Appendix A.10 Table 2, the proposed model achieves $1026 \pm 21$ on Walker Walk without distractions. This seems impossible, and undermines the credibility of the reported results.
- I find the paper a bit hard to follow. At the beginning of Section 3.1, I did not understand the relation between controllable representations and the conditional independence $I(A_{t-1}; S_t \mid S_t^+)=0$. Is this a necessary and/or sufficient condition and why? Also the PGM in Figure 2 may need more justification/explanation. Given that both $S_t^+$ and $S_t^-$ affect the reward and transition, is it reasonable to only learn $S_t^+$ and discard $S_t^-$?

**MINOR COMMENTS**
- The paper can be strengthened by comparing to TPC [1], a contrastive approach that outperforms DBC and CVRL.
- The total number of steps are different in Figure 5 and 7. It would be better to use the same number of steps for all tasks.
- What distractor levels are used in Figure 5 and 7?

[1] Temporal Predictive Coding For Model-Based Planning In Latent Space. Nguyen et al., ICML 2021.

**Summary Of The Paper:**

This paper tackles the problem of prioritizing functionally relevant information from complex observations for model-based RL. To this end, previous work has proposed to replace the reconstruction loss with contrastive loss. Building on that, this paper introduces an additional empowerment objective that is used for both representation learning and policy learning. Experiments show that the proposed model outperforms baselines on a set of DeepMind Control tasks with custom background distractions, which include other visually similar but uncontrollable agents. It is also shown that the similarity between learned states can match well to the similarity between groundtruth simulator states, according to the proposed metric.

**Summary Of The Review:**

I am leaning toward reject, due to potentially unfair comparisons and incorrect results.

---

My concerns have been adequately addressed during rebuttal, so I now recommend accept.

---

> ### Author Response · Authors · 2021-11-16
> **Author response (2/2)**
>
>
> - The average statistics reported in the last line of the abstract were based on Fig. 5 of the paper. However, we have removed the numbers from the updated paper, as we now have evaluations on two separate sets of distractor environments.
>
> - Thank you for helping us find a typo in Appendix A.10 Table 2 Walker Walk result in the submitted version. We have corrected the typo in the numbers in the table (which is now Table 4 in the Appendix of the revised paper).
>
> - Thank you for pointing us towards the TPC paper. We had missed it in our original submission, and now discuss it in the revised paper. However, we were unable to do an empirical comparison with this paper during the author response period, as the code is not publicly available, and it would take us some time to re-implement it. We will definitely do this, and incorporate the results in the revised paper.
>
> - We have now clarified in the paper that the distractors in Fig. 5 and 7 are of the same size as the observations (so level 3 as per the definition in section A.10 of the revised paper). We moved the study in section A.9 to the Appendix so as to incorporate the new results and details suggested by the reviewer, in the main paper.
>
> - The relationship for controllable representations is a necessary condition based on the PGM in Fig. 2. Controllable representations $S^+$ are the smallest subspace of $S$, $S^+ \leq S$, such that $I(A_{t-1};S_t|S^+_t)=0$. This is based on the intuition that controllable representations are the components that can be influenced by actions of the agent. Hence, given $S^+$, no further information about actions $A$ is present in state $S$. Regarding the second question - for capturing representations $S^-$ that are not controllable but do affect the rewards - this is done by the reward predictor model.
>
> - The figures are all with 2e6 environment interactions now, thank you for pointing this out.
>
> Kindly let us know if there is anything else that we can better clarify for the reviewer, for a revised assessment of our paper.

---

> ### Author Response · Authors · 2021-11-16
> **Author response (1/2)**
>
> We thank the reviewer for their detailed comments about our paper. We have updated the paper based on the comments, and highlighted the major updated parts in blue. In particular, we provided new comparisons with baselines on distractors from the Kinetics dataset that the reviewer suggested, added results for modified baselines with empowerment in policy learning, added details about the distractors and the baselines, and corrected the typo regarding a row of results in Table 4 (earlier Table 2) of the Appendix. Please find our detailed responses to the comments below
>
> - **Distractors.** The results in the paper were from distractors with the ILSVRC ( Imagenet large scale visual recognition challenge) dataset (Russakovsky et al., 2015). We have now clearly mentioned this in the revised section 4.1 of our paper with the reference. We used a total of 200 videos for training, and 50 videos for testing - where the testing videos were separate from those in training. We have provided these details in section A.6 of the Appendix.
>
> - **Tuning of baselines.** We tuned relevant hyperparameters of the baselines, and InfoPower through gridsearch. We referred to the respective hyperparameters mentioned in the appendices of the baseline papers, and the range of typical variations reported in the papers, for deciding the range for gridsearch. We have included details of this in the revised paper section A.7. For fair comparisons, we kept all the common hyper-parameter values the same as Dreamer. This was the protocol followed in prior works TIA and Ma et al. 2020. For the baseline TIA, we tuned the environment-specific parameters $\lambda_{R_{adv}}$ and $\lambda_{O_s}$ as mentioned in Table 2 of TIA. For $\lambda_{R_{adv}}$, we performed a gridsearch over the range $10k - 50k$ and for $\lambda_{O_s}$ we performed a gridsearch over the range $1-3$ to obtain the parameters for best performance, which we used for the plots in Fig 4 and Fig. 11. Respectively for walker-walk, cheetah-run, finger-spin, quadruped-walk, hopper-stand, ball-in-a-cup-catch, cartpole-balance-sparse, quadruped-run, the values of  $\lambda_{R_{adv}}$ are 30k,30k,20k,40k,30k,40k,20k,30k. The values of $\lambda_{O_s}$ are 2,2,1.5,2.5,2.5,2,2,2. For baseline DBC, we kept all the parameters the same as in Table 2 of the paper, because DBC does not have any environment-specific parameters and the same values were used for all environments in the DBC paper.  The DeepMDP agent and its hyperparameters are adapted from the implementation provided in the github repo of DBC.
>
> - **New results on the Kinetics dataset.** Thank you for suggesting the evaluations on greyscale distractors from the Kinetics dataset, for comparing directly to published results in the baseline papers. We have done this evaluation and reported the results for three environments in Table 1, where we compare directly with TIA and DBC results from the respective papers. From Table 1 we see that InfoPower slightly outperforms the baselines. We believe the performance gap is not as significant as Fig. 4 because the distractors being grayscale images, and not RGB, might be easier to ignore by the baselines as well as InfoPower.
>
> - **Regarding empowerment in the baselines.**  We want to emphasize that the incorporation of empowerment for visual model-based RL is a key contribution of our paper in learning controllable representations. Hence, we did not originally add empowerment to the baselines because that would effectively mean altering the baseline algorithms. However, based on the reviewer’s suggestion, we have now compared with modified versions of the baselines that include empowerment in policy learning, the same way as in InfoPower. The results for this are in Fig. 11 of the paper, where we see that the performance difference gap with InfoPower is lesser, indicating the utility of empowerment in policy learning for exploring controllable parts of the state-space. Kindly note that in this setting, we have effectively modified the baselines.

---

> ### Author Response · Authors · 2021-11-19
> **Request for discussion**
>
> Dear reviewer,
>
> Since the author response period is closing very soon, we wanted to send a gentle request for discussion. Please find our responses to the original review below, and the revised paper that we have uploaded. It will be very helpful if you could let us know whether we have addressed your concerns and if we can provide any additional clarification for a revised assessment of our paper.
>
> Thanks,
> Authors

---

> ### Comment · Reviewer_8ezz · 2021-11-19
> **Thanks and some additional questions**
>
> I would like to thank the authors for their detailed response. My concerns about the experiments have been largely addressed.
>
> I have some additional questions about the conditional independence:
> - From Figure 2, it seems that the environment transitions to state $S_t$ after action $A_t$ is applied. So why not let $I(A_t; S_t \mid S_t^+)=0$? Or why is it better to use $A_{t-1}$ instead of $A_t$? Can you also use any of the previous actions $A_{t-k}$?
> - You mentioned in your response that $S^-$ can be captured via reward prediction. Are you suggesting that the learned latent space is supposed to capture both $S^+$ and $S^-$? If so, would the empowerment loss and the reward prediction loss have contradictory effects?
> - Why should $S_{t+1}^-$ be independent of $S_t^+$ and $S_t^-$?

---

> > ### Author Response · Authors · 2021-11-20
> > **Thank you and response to the questions**
> >
> > Thank you so much for your response. We are happy to know that your concerns regarding experiments have been largely addressed. Please find our responses to the additional questions below:
> >
> > 1. Thank you for helping us spot a typo in the time-index of Fig. 2. The actions $A_t$ and $A_{t+1}$ should respectively be $A_{t-1}$ and $A_{t}$ to be consistent with our notation in the text. We apologize for this, which might have caused confusion. We have updated the paper with the modified figure 2. Since the state at time $t$ is influenced by $A_{t-1}$ as per the PGM, hence we have $I(A_{t-1};S_t|S_t^+)=0$ in the definition.
> >
> > 2. Yes, the learned latent space is indeed supposed to capture both $S^+$ and $S^-$, but kindly note that in most settings $S^-$ (that cannot be influenced by the agent, but affects rewards) is either a small part of $S$ or is absent. $S^+$ includes everything that is controllable by the agent (e.g. joints of the agent and position of the objects it interacts with) and $DS^-$ includes everything that is not controllable by the agent and that does not influence the rewards (e.g. background distractors) - these two are the major parts of $S$. The empowerment objective explicitly prioritizes capturing $S^+$ in the latent space as it is influenced by the actions of the agent. The empowerment objective cannot capture $S^-$ as it is not influenced by the agent's actions, and only influences the reward (Also, it does not de-prioritize capturing $S^-$ - it is just agnostic to it) . Hence, the reward predictor alone is responsible for capturing $S^-$ when it is present in the environment. So, in the case where both $S^+$ and $S^-$ are present, the two objectives - empowerment and reward predictor with help capture both these components in the latent space, without having contradictory outcomes.
> >
> > 3. Kindly note that $S_{t+1}$ should be independent of $S^+$ , because $S^+$ represents controllable parts of the state, and so if $S_{t+1}^-$ depends on $S_t^+$, then it would imply $S_{t+1}^-$ is controllable because $S_t^+$ is controllable (changing $S_t^+$ by the actions of the agent would change $S_{t+1}^-$). Regarding $S_{t}^-$, yes, $S_{t+1}^-$ need not be independent of $S_t^-$ , and could depend on it. We did not show this arrow in figure 2 to reduce clutter, and have updated the caption of figure 2 to reflect this. We would be happy to modify figure 2 and the caption further if the reviewer believes that we should make this dependence clear. Kindly let us know.
> >
> > Thank you once again for discussing with us. Please let us know if we can clarify the above points further, for a revised assessment of our paper.

---

### Official Review · Reviewer_Kbjg · 2021-11-03

**Correctness:** 4
**Technical Novelty And Significance:** 3
**Empirical Novelty And Significance:** 4
**Recommendation:** 8
**Confidence:** 4

**Main Review:**

Strengths:

I like the idea of the paper in that it appears to be logical and
almost obvious, after reading about it here. The solution elegantly
addresses the problems of representation, and policy learning, using
the same formalism. I also like the information-theoretic view as a general
formulation independent of a specific model.

Results appear to be very good on the chosen benchmarks. Results of
the ablation (appendix) and comparison without distractors is useful
to see, and the description of the distractors at least give a good
idea (but see below).


Weaknesses:

From the results table (in the main paper) it appears InfoPower comes
with much higher variance than other approaches. For results without
distractors (in the appendix), this is almost reversed. The variance
(and the differences) is not much discussed in the paper, but it may
be worthwhile to.

I cannot see the results being easily reproduced for a lack of detail
/ availability on the set of distractors, and was wondering if future
work and comparisons could be helped by data and/or code allowing
similar distractors as those used in the paper.


Other feedback:

While the approach and solution to the problem is new as far as I
know, relevant ideas have been discussed much earlier in a paper by S
Still, Information theoretic approach to interactive learning, EPL 85,
2009 https://arxiv.org/abs/0709.1948 that would be worth referencing.
I'd like to suggest including a brief discussion of the differences
and objectives of both ideas.

It might be useful to mention the relative difference in computation
of the approaches (on similar hardware). Compute times are given in
the paper and useful.

Typos etc
- p2: problem statement section: "captial" -> capital
- p4 bottom: "summmarise"
- In table 1, should the C-Dreamer L2 Rew@1M be bolded (because of the overlap / variances) (as in the level 1\
 case); the semantics of "bold" isn't explained so it's difficult to say.

**Summary Of The Paper:**

The paper introduces a method to learn representations for model-based
RL. The key idea is to find a representation that maximises
information between the action at the previous step and the current
latent representation, given the representation at the previous step,
thereby maximising the information that the representation encodes
about effects of an action. The work contributes a method and
evaluation on tasks where high-dimensional visual inputs are used to
control an agent in an environment with complex backgrounds as
distractors.



**Summary Of The Review:**

The paper is well written and motivated. I find the results compelling
and useful. There is not much to complain, except it would be good to
explore differences in variance of the different methods. Minor issues
are related to potential reproducibility / experimental setup, and a
reference to prior work.

---

> ### Author Response · Authors · 2021-11-16
> **Author response**
>
> We thank the reviewer for their helpful suggestions and positive comments about our paper. We have edited the paper based on reviewers’ comments. Kindly find our specific responses below
>
> - In our understanding the variance across random seeds is an issue for RL approaches in general, and is not very well analyzed. Hence, we evaluated the algorithms with 4 random seeds for training. We believe one of the reasons for the slightly lower variance of the baselines in the distractor setting is that there aren't any successful seeds, while for InfoPower there are more successful seeds and a few unsuccessful ones.
>
> - We will definitely release the code for the algorithms and the distractor environments. We have also added more details on the experimental settings in section A.7 of the Appendix, and included comparison to baselines that evaluated on an existing set of distractors, in Table 1 and compared directly to published results in those papers.
>
> - Thank you for pointing out the paper by S Still. It is definitely very relevant for our paper, and we have included a discussion of it in the related works.
>
> - We have included computation times of the baselines in the Appendix A.7 for comparison with InfoPower, on the same hardware.
>
>
> Kindly let us know if you have any suggestions about further improving the paper, and we would be very happy to incorporate them.

---

> > ### Comment · Reviewer_Kbjg · 2021-11-18
> > **Thanks**
> >
> > Reading over the other reviews, the responses and updates to the paper, my recommendation remains accept.

---

### Author Response · Authors · 2021-11-16
**Overview response for reviewers and ACs**

We thank the reviewers for their very helpful comments and feedback about our paper. All the reviewers appreciated the strong empirical results in the paper, and liked the intuitive and novel idea of the paper. They also agreed that the paper is targeting an important problem in vision based control. The reviewers pointed out specific concerns and points for improvement, which have helped us further improve the paper. Please find these changes (some parts are highlighted in blue for convenience) in the revised paper that we have uploaded. The summary of these changes are as follows:

1. We have included comparisons to prior approaches on distractors from the Kinetics dataset, suggested by reviewer 8ezz. This is in Table 1 of the revised paper. These results are on the exact setting reported by the two prior works TIA and DBC, and we compare directly to their published results.

2. We have included comparisons to modified baselines that incorporate empowerment in the policy learning objective, suggested by reviewer 8ezz. We would like to emphasize that the incorporation of empowerment for visual model-based RL is a key contribution of our paper in learning controllable representations. From these new results in Fig 11, we see that adding empowerment to policy learning of the baselines also helps improve their performance.

3. We have included details about the practical algorithm in section 3.4, improved Algorithm 1 for clarity, and referenced sections of the Appendix A.3, and A.7 that contain derivations for the optimization objective and details about the distractor settings, model architectures, and baselines.

4. We have edited the paper for typos, re-organized parts of the paper for clarity, and discussed some missing references based on the suggestions by the reviewers.

We would request the reviewers to kindly let us know if there is anything else we can clarify. Thank you for your time.

---

### Decision · Program_Chairs · 2022-01-20

**Decision:**

Accept (Poster)

**Comment:**

The reviewers agree that the proposed method to create a more robust representation of a task for model-based-RL is interesting and has significant merits. After some revision, more critical reviewers improved their ratings of the paper, such that there is unanimous agreement that the paper can be accepted to ICLR.